# Mapping (dis)agreement in hydrologic projections

Lieke Melsen[1], Nans Addor[2,3], Naoki Mizukami[2], Andrew Newman[2], Paul Torfs[1], Martyn Clark[2], Remko Uijlenhoet[1], and Adriaan Teuling[1]

[1]Hydrology and Quantitative Water Management Group, Wageningen University, The Netherlands
[2]National Center for Atmospheric Research (NCAR), Boulder, CO, USA
[3]Climatic Research Unit, University of East Anglia, Norwich, UK

*Correspondence to:* Lieke Melsen (lieke.melsen@wur.nl)

**Abstract.** Hydrologic projections are of vital socio-economic importance. Yet, they are also prone to uncertainty. In order to establish a meaningful range of storylines to support water managers in decision making, we need to reveal the relevant sources of uncertainty. Here, we systematically and extensively investigate uncertainty in hydrologic projections for 605 basins throughout the contiguous United States. We show that in the majority of the basins, the sign of change in average annual runoff and discharge timing for the period 2070–2100 compared to 1985–2008 differs among combinations of climate models, hydrologic models, and parameters. Mapping the results revealed that different sources of uncertainty dominate in different regions. Hydrologic model induced uncertainty on the sign of change in mean runoff was related to snow processes and aridity, whereas uncertainty in both mean runoff and discharge timing induced by the climate models was related to disagreement among the models regarding the change in precipitation. Overall, disagreement on the sign of change was more wide-spread for the mean runoff than for the discharge timing. The results demonstrate the need to define a wide range of quantitative hydrologic storylines, including parameter, hydrologic model, and climate model forcing uncertainty, to support water resources planning.

## 1 Introduction

A thorough understanding of the terrestrial component of the hydrological cycle is vitally important to ensure water resource management meets the many demands for water. Understanding the availability of water is important for domestic, agricultural and industrial consumers, including hydropower (Van Vliet et al., 2012) and inland navigation, and to design infrastructure to reduce the risk of flooding (Milly et al., 2002) and drought (Van Loon et al., 2016) in a changing climate. Models are needed to provide quantitative projections of how water resources will be affected by climate change. These projections remain, however, uncertain (Milly et al., 2005; Clark et al., 2016). To account for this uncertainty, quantitative hydrologic storylines, in which key features of climate change impact are represented, can guide water managers in developing dynamic policy pathways (Haasnoot et al., 2013). In order to establish a meaningful range of quantitative hydrologic storylines, we need to reveal, reduce and represent this uncertainty (Clark et al., 2016; McMillan et al., 2017).

All models in the Earth sciences are subject to conceptualization (Oreskes et al., 1994): Since not all of Nature's complexity can be captured at the scale of a model grid, processes are neglected, simplified, or scaled up in the model compared to reality.

A result of this conceptuality is that modellers make different decisions at different points in the model development process (Clark et al., 2011, 2015), producing models that have different portrayals of climate (Knutti and Sedláček, 2013; Knutti et al., 2013) or hydrologic (Mendoza et al., 2015a; Addor et al., 2014) change. Hydrologic projections require a long chain of models, with each step along the chain introducing uncertainty into the projection (Clark et al., 2016; Sonnenborg et al., 2015). As such,

long-term hydrologic projections are prone to the uncertainty in model inputs from downscaled General Circulation Model (GCM) projections (Knutti and Sedláček, 2013). But also the hydrologic model itself introduces uncertainty, both through the parameters (Vaze et al., 2010; Merz et al., 2011) and through the model structure, the conceptualization. The effect of model conceptualization on projected trends was underscored by a study on global trends in drought (Sheffield et al., 2012). Sheffield et al. (2012) compared the estimate of areas in agricultural drought obtained using the temperature-based Thornthwaite equation

to estimate potential evaporation, with the results obtained with the more physically founded Penman-Monteith equation. A much stronger (weaker) increase in global areas in drought was found when using the Thornthwaite (Penman-Monteith) formulation. The representation of underlying (physical) principles of hydrological processes in the hydrologic model can thus have a profound effect on the results and conclusion of a study.

Although uncertainty in hydrologic projections has already been discussed and investigated in literature, studies usually

focus on one source of uncertainty (Gutmann et al., 2014) or a limited number of catchments (Vidal et al., 2016; Addor et al., 2014; Dobler et al., 2012). Here, we investigate three sources of uncertainty (GCM forcing, hydrologic parameters, hydrologic model structure) in hydrologic projections for 605 basins throughout the contiguous United States over a wide range of climates, in order to reveal spatial patterns in uncertainty. We investigate the agreement in hydrologic projections for a volume metric (the annual average runoff) and a timing metric (the day of the year on which half of the discharge has passed).

**2  Methodology**

We assess the uncertainty in hydrologic projections using a multi-parameter multi-model multi-basin approach. We investigate the role hydrologic model parameters, hydrologic model, and GCM choice play in the uncertainty of hydrologic projections. We employ three frequently-used hydrologic models (SAC, VIC, HBV), constrained based on observations, and five different climate models from the Coupled Model Intercomparison Project Phase 5 (CMIP5, RCP8.5), to evaluate the sign of change in

a volume and a timing metric over the period 2070–2100 compared to 1985–2008. An overview of the procedure can be found in Figure 1.

The two investigated metrics are the long term mean runoff and the day of the year where half of the discharge has passed (referred to as 'discharge timing'). We focus on agreement in the sign of change only, and not on agreement in the size of the change. Although agreeing on the sign of change in mean runoff might already not be straightforward, it is a first and necessary

condition in robust projections. All other relevant variables, such as peak flows or drought (Roudier et al., 2016), are even harder to project, as they are related to runoff variability rather than mean runoff.

The large sample of basins employed in this study (605, spread over the contiguous United States, CONUS, Newman et al., 2014, 2015) provides the opportunity to study agreement in projections across a range of climate and catchment conditions,

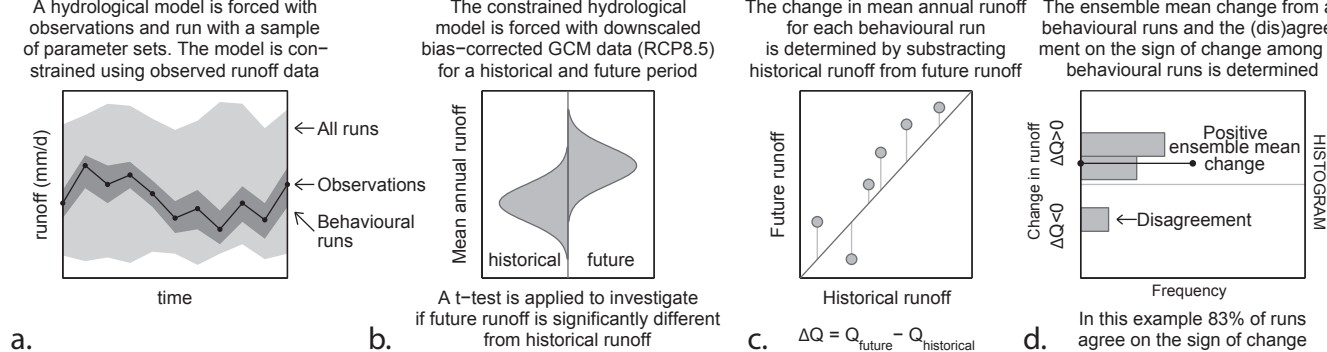

**Figure 1.** Overview of the conducted procedure, demonstrated for the mean runoff metric. For the discharge timing metric, the same procedure was employed. This procedure was repeated for fifteen combinations of three hydrologic models and five General Circulation Models (GCMs) using Representative Concentration Pathway (RCP) 8.5. a) The hydrologic model was run with a sample of parameter sets for the period 1985-2008. The model was forced with observed Daymet data (Thornton et al., 2012). The size of the parameter sample differed per model, from 1600 to 1900 (see Methods). Behavioural runs were identified based on 23 years of daily observed discharge data. b) The constrained model, i.e. the behavioural parameter sets, were forced with GCM data for the historical period 1985–2008 and future period 2070–2100. For each run, the mean annual runoff was determined for both periods. c) The change is defined as the difference in mean annual runoff between the historical and future period, and determined per run. d) Histogram showing the distribution of the change for the different model runs as shown in panel c. The sign of the ensemble mean change is determined, as well as the agreement among the different parameter sets on the sign of the ensemble mean change. The agreement is defined as the percentage of runs that project the same sign of change (positive change = increasing mean annual runoff, negative change = decreasing mean annual runoff) as the ensemble mean change. Finally, the sign of the ensemble mean change is compared for different combinations of hydrologic models and GCMs.

and to attribute the uncertainty to particular hydrological processes (Gupta et al., 2014). We will discuss the results based on an example for the VIC model, because this model is widely applied for climate impact assessments, and the CNRM-CM5 GCM was chosen as the reference GCM because this model has the lowest bias in CONUS (Sheffield et al., 2013). The results from the other hydrologic model - GCM combinations are presented in the appendices.

## 2.1 Catchment information

We will employ 605 basins throughout the contiguous United States. Input data and discharge observations for all basins are publicly available, see Newman et al. (2014, 2015). The median size of the catchments in the data set is 361 km², but areas range from 4 to 25,800 km². The mean elevation from the catchments ranges from 14 to 3640 m, with a median elevation of 454 m. The steepest catchment has a mean slope of 14.3°, the most gentle catchment a slope of 0.05°, and the median slope of the catchments in the data set is 1.5°. Land use in the catchments varies from mixed forest to grassland and from savannah to crop land. The publicly available CAMELS dataset (Addor et al., 2017a, b) contains several attributes of all basins: topographic characteristics, climate characteristics, hydrologic signatures, and land cover, soil and geological characteristics.

The catchments have been selected to minimize human influence, and are therefore mainly smaller headwater catchments (Newman et al., 2015). The discharge for all 605 basins is simulated for two periods. A historical period from 1980-2008 (of which we consider 23 years due to the 5 year model spin-up), and a future period from 2065-2100 (of which we consider 30 years due to the 5 year model spin-up). The different length of both simulation periods can influence the calculated means, but this effect is expected to be limited.

## 2.2 Hydrologic models and parameter sampling strategy

We apply three frequently used hydrologic models, which were run in a lumped fashion for the 605 basins: VIC 4.1.2h (Liang et al., 1994), SAC-SMA combined with SNOW-17 (Newman et al., 2015), and the TUWmodel following the structure of HBV (Parajka et al., 2007). A very brief description of the employed models can be found below.

### Variable Infiltration Capacity Model (VIC)

The Variable Infiltration Capacity (VIC Liang et al., 1994, 1996) model is a land-surface model that solves both the water and the energy balance. Three soil layers are distinguished. Evaporation takes place based on the available moisture from the upper soil layer. Dependent on the rooting depth water can be extracted from deeper soil layers for transpiration. For the upper two soil layers, the Xinanjiang formulation (Zhao et al., 1980) is used to describe infiltration. This formulation assumes that the infiltration capacity varies within an area. Moisture transport downwards is gravity-driven and only dictated by the moisture level of the upper soil layer. Runoff can consist of surface runoff and baseflow from the different soil layers. Surface runoff occurs when precipitation intensity exceeds the local infiltration capacity of the soil. Each soil layer can generate baseflow, linked to the soil moisture content of the layer based on the conceptualization of the Arno model (Francini and Pacciani, 1991). Generally, the deeper soil layers have a slower response. The snow model is a two-layer accumulation-ablation model, which solves both the energy- and the mass balance. At the top layer of the snow cover the energy exchange takes place. A zero energy flux boundary is assumed at the snow-ground interface.

### Sacramento Soil Moisture Accounting Model (SAC)

The Sacramento Soil Moisture Accounting model (SAC, Burnash et al., 1973; National Weather Service, 2002) is a bucket-type model that was developed by the US National Weather Service. The two basic components of SAC are tension water, water present in the soil but due to absorption to soil particles only removable through evaporation and transpiration, and free water, water that is available for percolation and drainage. Furthermore, SAC divides the soil into an upper and a lower zone. Runoff consists of direct runoff, interflow, and baseflow. Direct runoff is generated from impervious areas, and when rainfall intensity exceeds the infiltration rate of the soil or when the soil is saturated. Interflow is dictated by the 'free water' in the upper soil zone, whereas baseflow is determined based on the 'free water' in the lower soil zone. Snow-17 is an air-temperature-index-based snow accumulation and ablation model (Anderson, 1973).

**Hydrologiska Byråns Vattenbalansavdelning model (HBV)**

The Hydrologiska Byråns Vattenbalansavdelning Model (HBV, Bergström, 1976, 1992) was developed in Sweden. The HBV model consists of three main components: snow accumulation and melt (the snow routine), soil moisture (the soil routine), and response and river routing (response function and routing routine). Snow accumulation and snowmelt are parameterized with a degree-day expression, based on two parameters, one which represents the threshold level above which snowmelt starts and one below which precipitation falls as snow. The soil routine is controlled by the maximum soil moisture storage, a non-linear function that describes the relation between soil moisture level and recharge and a parameter that links the soil moisture level to evaporation. Runoff is generated by recharge from the soil moisture routine into two reservoirs: a fast responding reservoir generating fast response runoff, and a slow responding reservoir (baseflow) that is fed with percolation water from the fast responding reservoir.

For each model, a representative set of parameters that capture the essential hydro-climatological features was identified. A large range of parameters was sampled using a Sobol'-based Latin Hypercube sample: 17 parameters for VIC (Demaria et al., 2007; Chaney et al., 2015; Melsen et al., 2016; Mendoza et al., 2015b), 18 parameters for SAC (Newman et al., 2015; Lhomme, 1997), and 15 for HBV(Parajka et al., 2007; Uhlenbrook et al., 1999; Abebe et al., 2010). Physically realistic parameter boundaries were determined, based on the literature: see Appendix Tables C1, C2, and C3. First, based on the average parameter values (halfway between the maximum and minimum value as mentioned in the Appendix Tables), 100 base runs were sampled. Subsequently, each parameter was sampled 100 times by applying perturbations to the base runs. This implies that for each of the 605 basins, SAC was run 1900 times, VIC 1800 times, and HBV 1600 times (leading to a total of 3.2 million runs per time period). The hydrologic models were forced with daily Daymet observed meteorological variables (Thornton et al., 2012) and the model output from the different parameter sets was compared with daily USGS observed discharges over a 23 year period (1985–2008). The period 1980–1985 was used as spin-up for the model. Five years of spin-up was considered sufficient: For VIC, when run on an hourly basis, three months was shown to be sufficient to diminish the effect of initial conditions (Melsen et al., 2016), and Seibert and Vis (2012) state that one year warm-up period on a daily basis is sufficient in most cases for the HBV model. The effect of initial conditions can be longer in drier climates. The parameter sets were considered behavioural as soon as they fulfilled a criterion that minimizes the Euclidean distance between observations and simulations for three components: the correlation, the relative variability, and the relative bias, the Kling-Gupta Efficiency ($E_{KG}$, Gupta et al., 2009).

$$E_{KG}(Q) = 1 - \sqrt{(r-1)^2 + (\alpha - 1)^2 + (\beta - 1)^2}, \tag{1}$$

where $r$ is the correlation between observed and simulated runoff, $\alpha$ is the standard deviation of the simulated runoff divided by the standard deviation of observed runoff, and $\beta$ is the mean of the simulated runoff, divided by the mean of the observed runoff. The parameter set needed to result in a $E_{KG}$ of at least 0.5 on a daily basis over 23 years in order to be considered behavioural, see Figure 1a. If none of the parameter sets fulfilled the performance criterion, the hydrologic model was considered non-

behavioural. Using a fixed threshold can result in a different number of parameter sets being classified as behavioural for the different catchments. Appendix Figure C1 displays the percentage of behavioural parameter sets per model per catchment.

## 2.3 GCM forcing data

The constrained hydrologic models were forced with statistically downscaled and bias corrected GCM output using Bias Corrected Spatial Disaggregation (Wood et al., 2004) for a historical (1980–2008) and future (2065–2100) period and run with a daily time step (Figure 1b). The first five years of both periods were used as spin-up period and ignored in the analysis. Five different climate models from the Coupled Model Intercomparison Project Phase 5 (CMIP5) using Representative Concentration Pathway 8.5 (RCP8.5) were employed: CNRM-CM5, IPSL-CM5A-MR, CCSM4, MPI-ESM-MR, and INM-CM4. These five climate models were selected based on the climate model genealogy proposed by Knutti et al. (2013). They defined GCM families based on their predicted change in temperature and precipitation for the end of the 21st century using RCP8.5. By selecting one member (GCM) of each GCM family, we approach the full range of projections by all GCMs. From each family, the member with the smallest bias in temperature and precipitation for the contiguous United States (Sheffield et al., 2013) was selected.

## 2.4 Test for significant change

In order to test if the projected change in mean annual runoff and discharge timing was significant, a t-test was applied, which compared the distribution of the metrics over the behavioural parameter sets for the historical period with the distribution of the metrics for the same parameter sets for the future period (Figure 1b). The threshold in order to be qualified as significant was $p<0.05$. In order to apply the t-test, we set a pragmatic lower boundary of at least 10 parameter sets that needed to be behavioural. For VIC, SAC, and HBV, 0.5, 0.8, and 0.7% of the basins, respectively, had less than 10 but at least 1 behavioural parameter sets. In these basins, the significance of the projected change could not be tested. We first considered excluding the basins with a non-significant change from the analysis, following the approach of Knutti and Sedláček (2013) where agreement and significance are combined in a robustness metric. However, it turned out that none of the basins experienced a consistent non-significant change in any of the two metrics over different hydrologic models and GCMs. Therefore none of the basins was excluded.

## 2.5 Analysis of agreement in projected change

To determine the change in the two investigated metrics for a basin, the simulated mean annual runoff and the discharge timing over the period 1985–2008 were compared with the simulated mean annual runoff and discharge timing over the period 2070–2100 for each behavioural parameter set. The difference in the metrics between both periods is the projected *change*, where the change in mean annual runoff can be an *increase* or *decrease* (Figure 1c), and the change in discharge timing can be *earlier* or *later* in the year. The ensemble mean change was then determined as the mean change for all the behavioural parameter sets (Figure 1d). The ensemble mean change has been determined for each hydrologic model and climate model combination

separately. In this study we particularly focus on the sign of the ensemble mean change (i.e., an increase or decrease in mean annual runoff, and an earlier or later discharge timing).

We related the sign of the ensemble mean change in both metrics to three sources of uncertainty: hydrologic model parameters, the choice of the hydrologic model, and climate forcing. To identify the uncertainty induced by the representative parameter sets (the behavioural runs), the agreement among the representative sample of parameter sets on the sign of the change was determined per basin (Figure 1d). To investigate the effect of the choice of the hydrologic model, the sign of the ensemble mean change projected for the three different hydrologic models was compared per basin. To investigate the impact of the type of climate forcing on hydrologic projections, the sign of the ensemble mean change obtained by forcing the same hydrologic model with five different GCM outputs was compared.

Given the spatial patterns observed in the (dis)agreement when comparing hydrological models and GCMs, we investigated if catchment and climate characteristics influence the (dis)agreement. To assess the influence on the hydrological model agreement, we divided all basins into three categories: basins in which the three hydrologic models agree on the sign of the change, basins where the three hydrologic models disagree, and basins where the three hydrologic models are non-behavioural. For all basins, eight catchment and climate characteristics were identified. With a t-test, the climate and catchment characteristics for each category were compared to the complete sample of basins, to identify which characteristics were significantly ($p<0.05$) different among the three categories. To investigate the effect of the GCM, the sign of the ensemble mean change from the same hydrologic model forced with five different GCMs, was compared. In this case, the basins were divided in two categories: basins where the model outputs obtained with different GCM forcings consistently agree on the sign of change, and basins where the model outputs disagrees using different GCMs. Subsequently, these categories have been related to four climate change characteristics and tested on significance ($p<0.05$) using a t-test.

## 2.6 Combined sources of uncertainty

To investigate if there is any correlation between the three investigated sources of uncertainty, we combined the results of the previous analyses, where the three sources of uncertainty were investigated separately. For each basin and for all model combinations, we determined if hydrologic parameters (fifteen GCM - hydrologic model combinations), hydrologic model (five GCM - hydrologic model combinations) and/or GCM (three GCM - hydrologic model combinations) lead to disagreement in the projections. This resulted in in a classification of each basin into eight different classes; either only one of the three sources of uncertainty lead to disagreement in the projection (parameters or hydrologic model or GCM), two different sources of uncertainty lead to disagreement (e.g. both the parameters and the hydrologic model), all three different sources of uncertainty lead to disagreement, or none of the sources of uncertainty lead to disagreement. To visualize these results, we determined a grid-based ($1\times1°$) maximum likelihood: for each grid cell, the most frequent classification from the basins within that grid cell was determined.

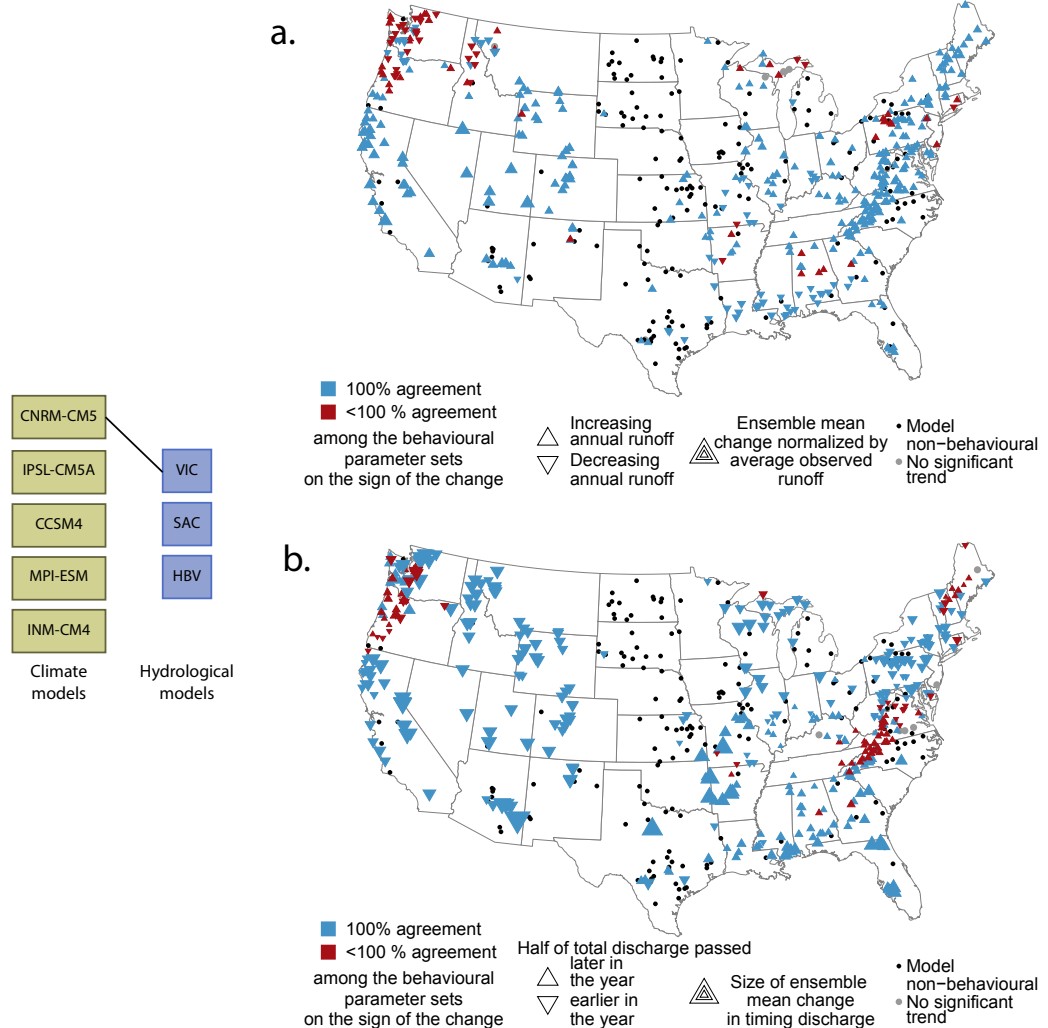

**Figure 2.** Distribution of uncertainty in the sign of change in mean runoff (a) and discharge timing (b) over the contiguous United States when different behavioural parameter sets are compared. Left: Combination of climate model and hydrologic model for which the results are displayed. In this case, the CNRM-CM5 climate model with the Variable Infiltration Capacity (VIC) hydrological model. (SAC=Sacramento Soil Moisture Accounting Model, HBV=Hydrologiska Byråns Vattenbalansavdelning Model). a) The agreement among the different model runs (representing different behavioural parameter sets) of the VIC model on the sign of the ensemble mean change in mean annual runoff. b) The agreement among the different model runs of the VIC model on the sign of the ensemble mean change in discharge timing. The direction of the triangle-marker shows the sign of the ensemble mean change, the size of the marker indicates the relative projected change.

## 3   Results

### 3.1   Impact of behavioural parameters

#### 3.1.1   Mean runoff

Results show that the sign of the change in mean annual runoff can be affected by the choice of hydrologic model parameters,
even when only considering hydrologic model parameter sets that provide reasonable simulations of current hydrological
behaviour (see Section 2.2). As an example for the VIC model forced with CNRM-CM5, we find a disagreement on the sign
of the change in 11% of the basins. In these basins, some parameter sets lead to an increase in mean annual runoff, while
other parameter sets lead to a decrease in mean annual runoff in the future under the same forcing. Figure 2a shows the spatial
distribution of these basins. Clustering of basins with disagreement, especially in the north-west, is visible. In the majority of
the basins (59%) there is unanimous agreement on the sign of the change. However, in 29% of the basins no representative
parameter sets could be identified, i.e., the hydrologic model could not capture the hydrological behaviour of the basin in
current climate with observed forcing, and is therefore qualified as non-behavioural. In 1% of the basins the change was not
significant, or less than 10 parameter sets were identified as representative (see Section 2.2). The percentage of basins in
which there is unanimous agreement on the sign of change among the different behavioural parameter set runs depends on the
employed hydrologic model and the GCM (see Appendix Figure A1). On average, VIC leads to 100%-agreement in 55% of
the basins (averaged over five GCMs), followed by SAC (46%), and HBV (43%).

#### 3.1.2   Discharge timing

Also for the timing metric, behavioural hydrologic model parameters can lead to disagreement on the sign of change. The
results for VIC forced with CNRM-CM5, shown in Figure 2b, have a disagreement in the sign of change in 15% of the basins
(55% unanimous agreement, 29% no behavioural runs). Spatially, we recognize two clusters of disagreement: one in the north-
west, although in different basins than where disagreement in the mean runoff metric was found, and one close to the central
east coast in the Appalachen region, which was not found for the mean runoff metric. On average over all model combinations,
hydrologic parameters lead to disagreement in discharge timing in 19% of the basins, with SAC being most sensitive for dis-
agreement in the sign of change (on average in 27% of the basins), see also Appendix Figure B1.

A more stringent criterion to identify representative parameter sets can potentially decrease the disagreement introduced
by the parameters, but at the same time increases the number of basins in which the models are non-behavioural. Inherent
to our approach is the relatively coarse parameter sample, which can explain why some models are non-behavioural in some
basins while other studies have applied the same model in comparable regions (e.g. Beck et al., 2016). A larger parameter
sample might decrease the number of non-behavioural basins and even allow for a more stringent selection criterion. Another
reason for the non-behavioural basins could be that the warm-up period as applied in this study (five years) was not completely
sufficient for drier climates. Beck et al. (2016), for example, applied a warm-up period of at least ten years. Furthermore, it is

important to remark that the parameters have been classified based on a general performance metric, not specified for the mean runoff and/or discharge timing that were evaluated here. Not including the evaluation metrics in the optimization can lead to an inadequate depiction of those metrics (e.g. Pool et al., 2017). Including mean runoff and discharge timing as metrics in the parameter classification procedure may decrease the disagreement, although this will, of course, also decrease the general
performance as expressed in the Kling-Gupta efficiency. Another approach would be to constrain the hydrologic models on observations of several different hydrological states and fluxes, such as soil moisture and ground water (Koster et al., 2010; Rakovec et al., 2016).

## 3.2 Impact of hydrologic model

### 3.2.1 Mean runoff

The sign of the change in mean annual runoff is affected by the choice of the hydrologic model. The choice of the hydrologic model leads to disagreement in 26% of the basins when the three models are forced with the same CNRM-CM5 output. Figure 3a shows the basins in which the three employed hydrologic models (dis-)agree on the sign of the change in mean annual runoff. The east-coast shows the clearest boundary: in the north-east the models disagree, whereas in the south-east the models agree. In 46% of the basins, three (or two, if one model was non-behavioural) hydrologic models agreed on the sign of the change.
In 14% of the basins two hydrologic models were non-behavioural, such that a mutual model comparison was impossible. In the remaining 14% of the basins, none of the employed hydrologic models were behavioural. The agreement among the hydrologic models varied when forced with different GCMs (see Appendix Figure A2). With CCSM4, the highest agreement among hydrologic models was established (in 59% of the basins), with INM-CM4 the lowest agreement (45%). The spatial structure of the agreement among the hydrologic models, as demonstrated in Figure 3a, suggests a link with catchment and
climate characteristics. We compared several characteristics for three different agreement categories (agreement, dark and light blue dots in Figure 3a; no agreement, orange and red dots in Figure 3a; three models non-behavioural, black dots in Figure 3a). Figure 4 shows the spatial distribution and summarizes the characteristics of the basins within each category. We only discuss the characteristics which were consistently significantly different when the three hydrologic models were forced with different GCMs.

The basins in which the three hydrologic models agree on the sign of the change (Figure 4a) experience a significantly lower projected temperature change compared to all other basins. Furthermore, these basins have on average a lower aridity, less or no dry periods, and a lower snow-day ratio. A logical explanation, namely that the basins in which the models agree experience a significantly larger change in precipitation as demonstrated in Figure 4b, was found not to be consistent among the five GCMs. The basins in which the hydrologic models disagree on the sign of the change (Figure 4c) are characterized by
a consistently higher slope and elevation, and related to that, a significantly higher snow-day ratio (Figure 4d). Disagreement among the hydrologic models can thus be attributed to the conceptualization of snow accumulation and melt processes. The basins in which none of the hydrologic models was able to capture current hydrological behaviour with observed forcing (i.e., where the models were non-behavioural, Figure 4e) have a significantly higher aridity and intermittent stream flow behaviour

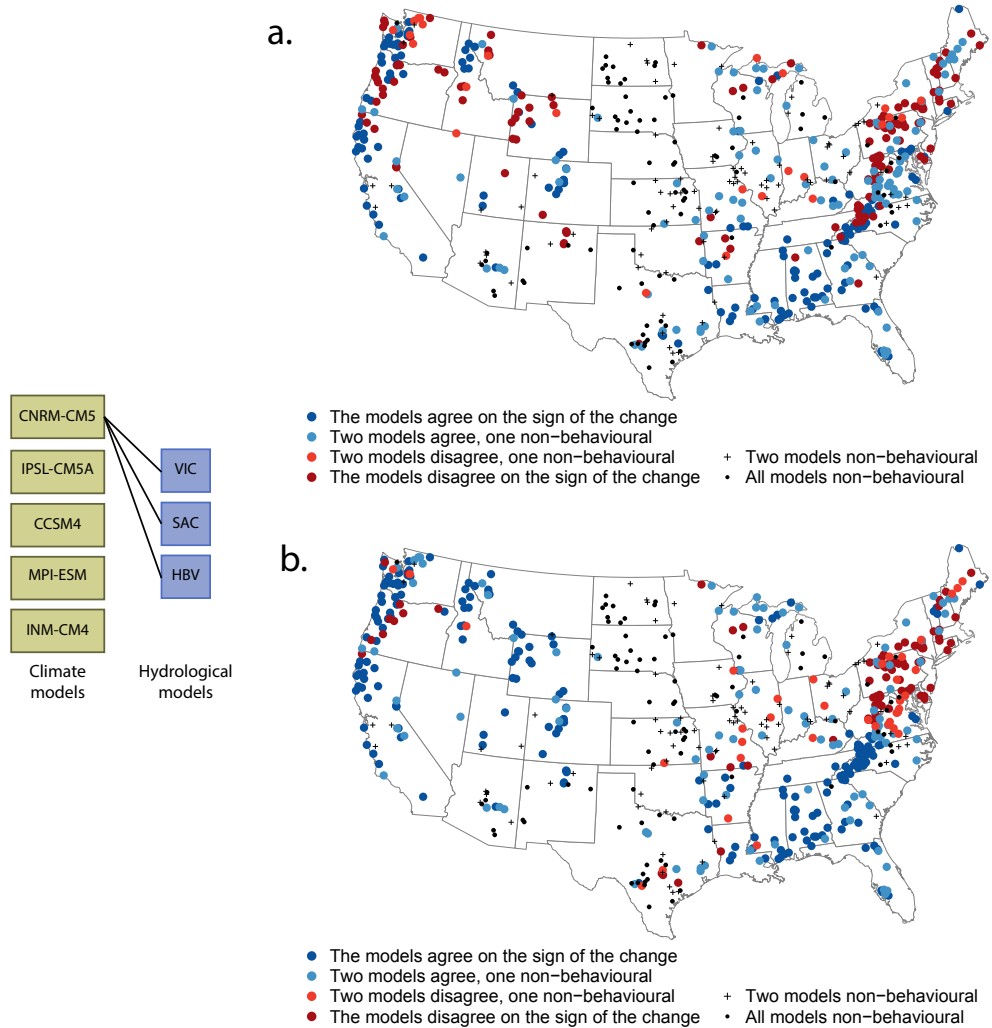

**Figure 3.** Distribution of uncertainty in the sign of change in mean runoff (a) and discharge timing (b) over the contiguous United States when three hydrological models are compared. Left: Combination of climate model and hydrologic model for which the results are displayed. VIC = Variable Infiltration Capacity Model, SAC=Sacramento Soil Moisture Accounting Model, HBV=Hydrologiska Byråns Vattenbalansavdelning Model. a) Agreement among the three different hydrologic models (all forced with CNRM-CM5) on the sign of the ensemble mean change in mean annual runoff. b) Agreement among the three different hydrologic models (all forced with CNRM-CM5) on the sign of the ensemble mean change in discharge timing.

(no-flow periods). This is related to basins with a larger area and a lower slope (Figure 4f). These results imply that all hydrologic models have difficulty in mimicking dry conditions, where the interplay between soil moisture and evapotranspiration becomes important (Seneviratne et al., 2010). Our lack of understanding in these processes is of concern, particularly as arid-

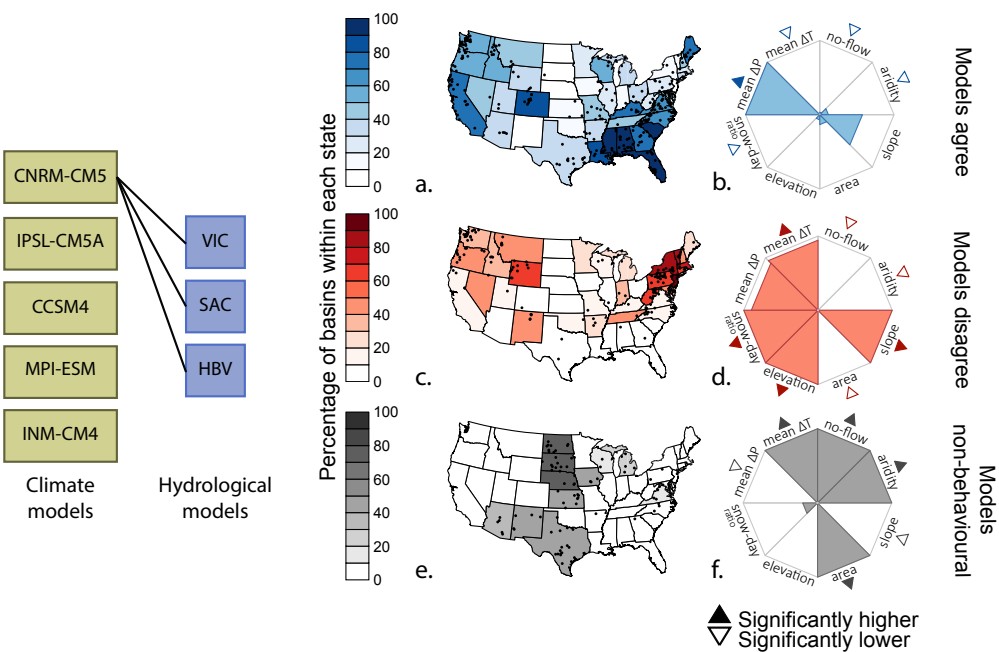

**Figure 4.** Spatial distribution of the basins in which the three hydrologic models agree on the sign of change in mean annual runoff (a), disagree on the sign (c) and of the basins in which all model were non-behavioural for current climate with observed forcing (e). The rose plots (b, d and f) show the standardized catchment and climate characteristics for catchments shown in panels a), c) and e), respectively. In the rose plots, 'mean ΔP' stands for the mean change in precipitation, and 'mean ΔT' for the mean change in temperature. The hydrologic models (VIC = Variable Infiltration Capacity Model, SAC=Sacramento Soil Moisture Accounting Model, HBV=Hydrologiska Byråns Vattenbalansavdelning Model) were forced with CNRM-CM5 projections (Representative Concentration Pathway 8.5). The characteristics in the rose plots have been standardized from 0 to 1, where 0 represents the lowest value of the characteristic for the three displayed groups, and 1 the highest value of the characteristics for the three displayed groups.

ity is expected to increase in the future (Berg et al., 2016). Improved representation of these processes is needed, which will probably entail scrutinizing soil moisture - evapotranspiration feedbacks (Roderick et al., 2015). For the basins in Figure 4e, a significantly lower change in precipitation is projected, consistent among the five GCMs.

### 3.2.2 Discharge timing

5    The sign of change in discharge timing can also be affected by the choice of the hydrological model. In 19% of the basins, the choice of the hydrologic model leads to disagreement in the sign of change when forced with the same CNRM-CM5 output, as shown in Figure 3b. For the displayed case, in 52% of the basins, three (or two, if one model was non-behavioural) hydrologic models agreed on the sign of the change. On average over all the different climate models, the choice of the hydrologic model structure leads to disagreement in 23% of the basins (see also Appendix Figure B2). As with the mean runoff metric, we also

10   recognize a spatial structure in the agreement on the sign of change in discharge timing in Figure 3b. Also for the discharge

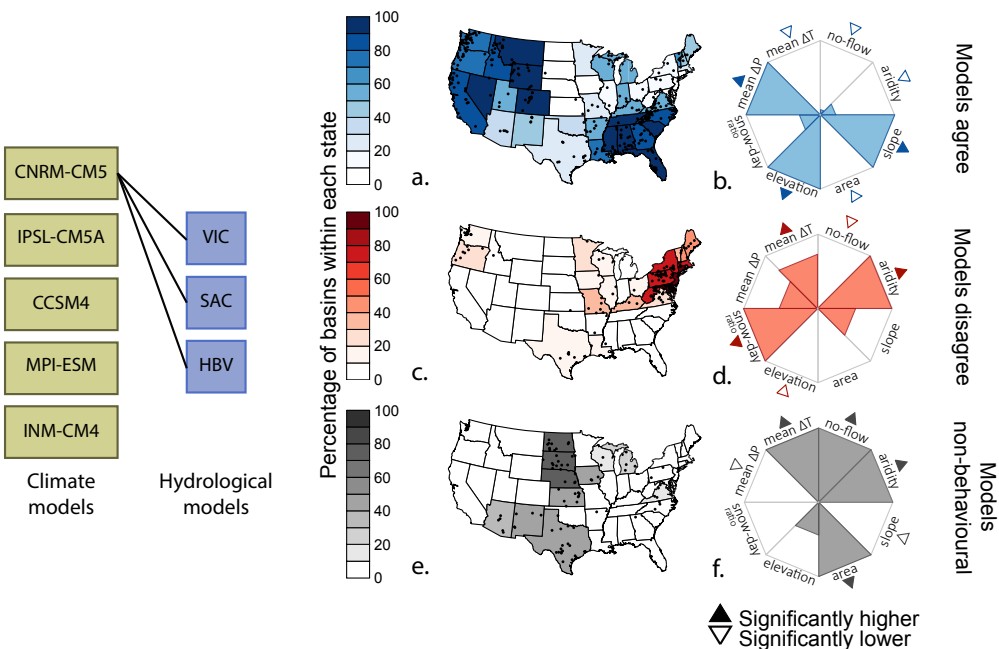

**Figure 5.** Spatial distribution of the basins in which the three hydrologic models agree on the sign of change in discharge timing (a), disagree on the sign (c) and of the basins in which all model were non-behavioural for current climate with observed forcing (e). The rose plots (b, d and f) show the standardized catchment and climate characteristics for catchments shown in panels a), c) and e), respectively. In the rose plots, 'mean ΔP' stands for the mean change in precipitation, and 'mean ΔT' for the mean change in temperature. The hydrologic models (VIC = Variable Infiltration Capacity Model, SAC=Sacramento Soil Moisture Accounting Model, HBV=Hydrologiska Byråns Vattenbal-ansavdelning Model) were forced with CNRM-CM5 projections (Representative Concentration Pathway 8.5). The characteristics in the rose plots have been standardized from 0 to 1, where 0 represents the lowest value of the characteristic for the three displayed groups, and 1 the highest value of the characteristics for the three displayed groups.

timing metric, we compared several characteristics for three different agreement categories (agreement, dark and light blue dots in Figure 3b; no agreement, orange and red dots in Figure 3b; three models non-behavioural, black dots in Figure 3b). The spatial distribution and the summarized characteristics are displayed in Figure 5. Only the characteristics which were consistently significantly different over the different GCMs are discussed.

5      Three catchment characteristics are consistently significantly different for the basins in which there is agreement among hydrological models on the sign of change in discharge timing (Figure 5a). These basins are characterized by a significantly lower amount of no-flow periods, and these basins have a higher slope and elevation compared to the basins where hydrological models lead to disagreement (Figure 5c) or where to models are non-behavioural (Figure 5e). The results seem to imply that the models agree on discharge timing in mountainous regions. The basins in which the hydrologic models disagree on the sign of the change in discharge timing (Figure 5c) have fewer no-flow periods, a higher aridity, and a lower elevation compared to

10  all other basins. In contrast to the results for the mean runoff metric, mean ΔT and snow day ratio do not consistently appear as

explanatory variable, although especially the latter could potentially influence discharge timing. The basins in which none of the hydrologic models was able to capture current hydrological behaviour with observed forcing are, naturally, equal for both the volume and the timing metric (Figure 5e), and thus characterized by higher aridity, more no-flow periods, a lower slope, a larger area, and a smaller projected change in precipitation, as discussed in Section 3.2.1.

## 3.3    Impact of model forcing

### 3.3.1    Mean runoff

Uncertainty in the sign of change introduced by the climate forcing is shown to have impact in the majority of the basins: In 60% of the basins a different GCM forcing leads to a different sign of change in mean runoff when the VIC hydrologic model is employed. Figure 6a shows that only in 11% of the basins, the sign of the change is consistent when the same hydrologic model (VIC) is forced with different climate model outputs. In the remaining 29% of the basins, the hydrologic model was non-behavioural. When HBV is applied with five GCMs, 16% of the basins show a consistent sign of change, for SAC only 8% (see Appendix Figure A3). Climate change characteristics have a major influence on the (dis)agreement in the sign of the change when the hydrologic model is forced with five different GCMs. The spatial distribution and the related climate change characteristics of two different categories (agreement, dark blue dots in Figure 6a; disagreement, orange and red dots in Figure 6a) are shown in Figure 7. We only discuss the characteristics which were consistently significantly different among the three different hydrologic models when forced with five GCMs.

The basins with agreement on the sign of the change (Figure 7a) are characterized by a significantly lower standard deviation in the projected change in precipitation (Figure 7b), i.e., the GCMs agree more on the projected change in precipitation. The basins in which the models disagree on the sign of the change (Figure 7c) have a significantly larger change in precipitation, a smaller change in temperature, a smaller standard deviation in temperature among the five GCMs, but a larger standard deviation in precipitation among the five GCMs (Figure 7d).

### 3.3.2    Discharge timing

In 36% of the basins, different GCM forcing leads to a different sign of change in discharge timing, when the VIC model is employed (Figure 6b), whereas in 35% of the basins, there is agreement on the sign of change when the same hydrological model is forced with five different GCM outputs. This is substantially different from the findings for the mean runoff metric (where 60% and 11% of the basins experienced disagreement and agreement, respectively). Also for the other hydrological models, the percentage of basins in which there is agreement on the sign of change is higher for the discharge timing than for the mean runoff: with SAC, in 31% of the basins there is agreement on the sign of change, with HBV, 24% of the basins (see Appendix Figure B3), although the impact of the GCMs on the agreement is still remarkable. Figure 6 shows that the different results for the two metrics can particularly be found in the west part of the US. The spatial distribution of (dis)agreement

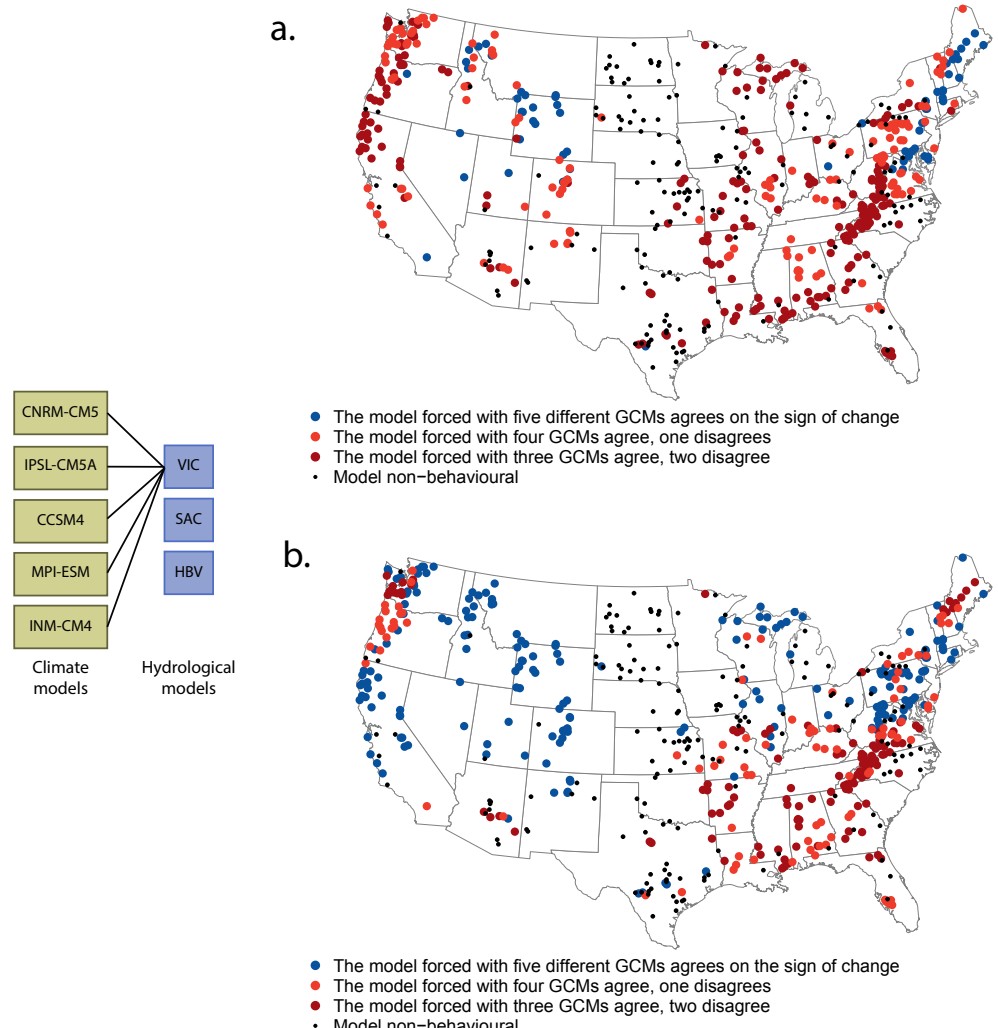

**Figure 6.** Distribution of uncertainty in the sign of change over the contiguous United States when different General Circulation Models (GCMs) are used to force a hydrological model. Left: Combination of climate model and hydrologic model (VIC = Variable Infiltration Capacity Model, SAC = Sacramento Soil Moisture Accounting Model, HBV = Hydrologiska Byråns Vattenbalansavdelning Model) for which the results are displayed. Right: Agreement on the sign of the ensemble mean change when the same hydrologic model (VIC) is forced with data from five different GCMs.

on the sign of change in discharge timing is related to climate change characteristics in Figure 8. Only the climate change characteristics that have shown to consistently significantly differ over the three hydrological models are discussed.

The basins in which there is agreement on the sign of change in discharge timing (Figure 8a) experience a significantly higher change in temperature and change in precipitation (Figure 8b). Disagreement among the GCMs concerning the change

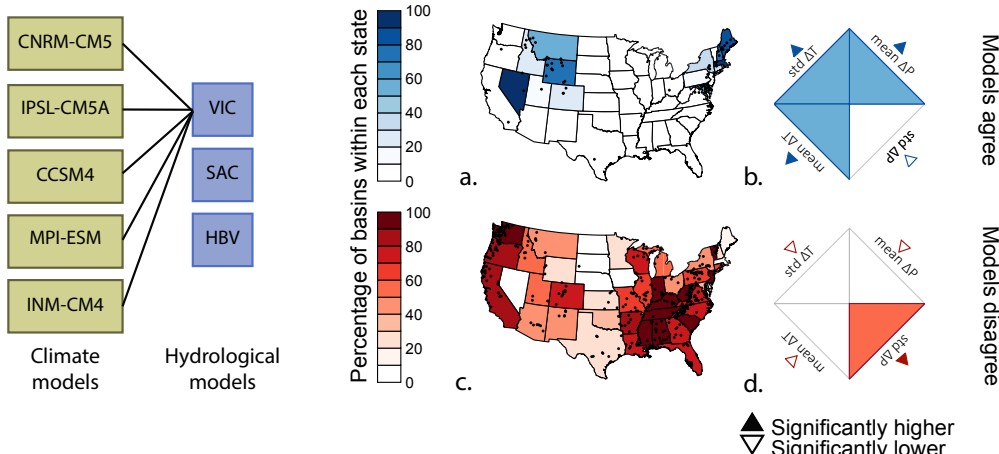

**Figure 7.** Spatial distribution of the basins in which Variable Infiltration Capacity Model (VIC) simulations driven by the five climate models fully agree on the sign of change in mean annual runoff (a), and disagree on the sign of change (c). The rose plots (b, and d) show standardized climate change characteristics for catchments shown in panels a) and c), respectively. The 'std∆T' (P) stands for the standard deviation of the projected change in Temperature (Precipitation) for the five different climate models, 'mean ∆T' (P) is the mean projected change in Temperature (Precipitation) of the five different climate models. The characteristics in the rose plots have been standardized from 0 to 1, where 0 represents the lowest value of the characteristic for the two displayed groups, and 1 the highest value of the characteristics for the two displayed groups. Left: Combination of climate model and hydrologic model (VIC = Variable Infiltration Capacity Model, SAC = Sacramento Soil Moisture Accounting Model, HBV = Hydrologiska Byråns Vattenbalansavdelning Model) for which the results are displayed.

in temperature (std ∆T) was not consistently significantly higher for the three different hydrological models. For all three hydrological models, the basins in which different GCM forcing leads to disagreement on the trend direction (Figure 8c) experience a lower change in temperature, a higher change in precipitation, a higher agreement on the change in temperature, but, as we also saw for the mean runoff metric, a higher disagreement on the change in precipitation. These results show that precipitation is the driving force for hydrologic models, and that the disagreement among GCMs regarding the change in precipitation introduces uncertainty in the hydrologic projection. Reducing this uncertainty, however, is not a trivial task (Knutti and Sedláček, 2013; Fatichi et al., 2016). While it may seem illogical that both the basins with agreement (Figure 7a, 8a) and the basins with disagreement (Figure 7c, 8c) have a significantly higher mean change in projected precipitation (panels b and d in Figures 7 and 8), this is because the basins in each category have been compared to all other basins, including the basins in which the hydrologic model was non-behavioural.

### 3.4 Combined uncertainty

One could expect that uncertainty introduced by the parameters is related to uncertainty introduced by the choice of the hydrologic model. This is, however, not the case. Figure 9 provides an overview of the investigated sources of uncertainty and their combined spatial distribution (see also Appendix Figure A4 and B4).

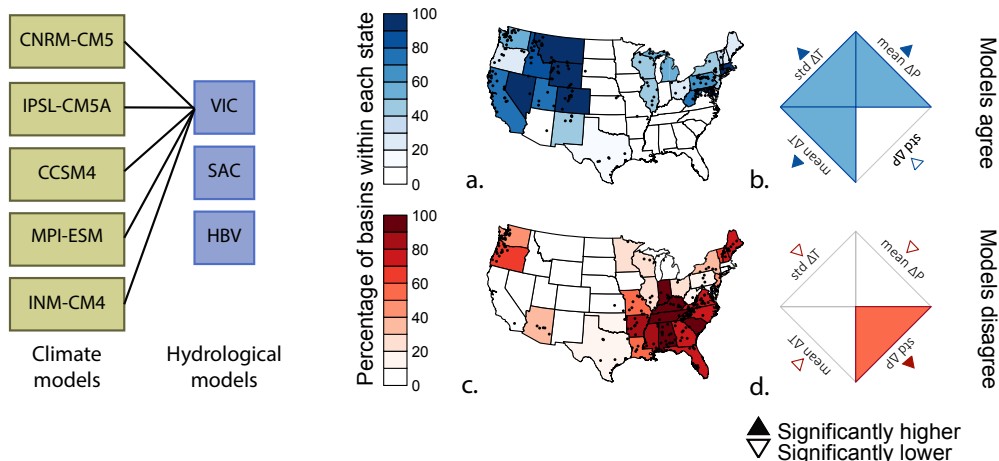

**Figure 8.** Spatial distribution of the basins in which Variable Infiltration Capacity Model (VIC) simulations driven by the five climate models agrees on the sign of change in discharge timing (a), and disagree on the sign of change (c). The rose plots (b, and d) show standardized climate change characteristics for catchments shown in panels a) and c), respectively. The 'std$\Delta$T' (P) stands for the standard deviation of the projected change in Temperature (Precipitation) for the five different climate models, 'mean $\Delta$T' (P) is the mean projected change in Temperature (Precipitation) of the five different climate models. The characteristics in the rose plots have been standardized from 0 to 1, where 0 represents the lowest value of the characteristic for the two displayed groups, and 1 the highest value of the characteristics for the two displayed groups. Left: Combination of climate model and hydrologic model (VIC = Variable Infiltration Capacity Model, SAC = Sacramento Soil Moisture Accounting Model, HBV = Hydrologiska Byråns Vattenbalansavdelning Model) for which the results are displayed.

### 3.4.1 Mean runoff

For the mean runoff metric, depending on which combination of hydrologic model and GCM is employed, 0 to only 5% of the basins experience both uncertainty from the parameter sets and from the hydrologic model choice, and not from the climate model. The combination hydrologic model - GCM is more frequent (3 - 14%) and the combination of parameter sets and GCM is most frequent (3 - 19%). In 1 to 16% of the basins, all three factors lead to uncertainty. The map in Figure 9a also reveals a spatial pattern in relevant uncertainties; in the north-west combinations of parameter sets and climate model can be found, whereas in the north-east the combination hydrologic model - GCM is more common. In the south-east only GCM choice is the main source of uncertainty. The models are non-behavioural in the great plains area.

### 3.4.2 Discharge timing

Also for the discharge timing metric, only 0 to 5% of the basins experience uncertainty from both the parameter sets and the hydrologic model choice. More frequent is the combination parameter sets - GCM uncertainty (2 - 10%), most frequent is the combination hydrologic model - GCM (3 - 12%). Figure 9b shows spatial clustering of the different sources of uncertainty: in the north-west, all three investigated sources of uncertainty influence the sign of the change in discharge timing, in the south-

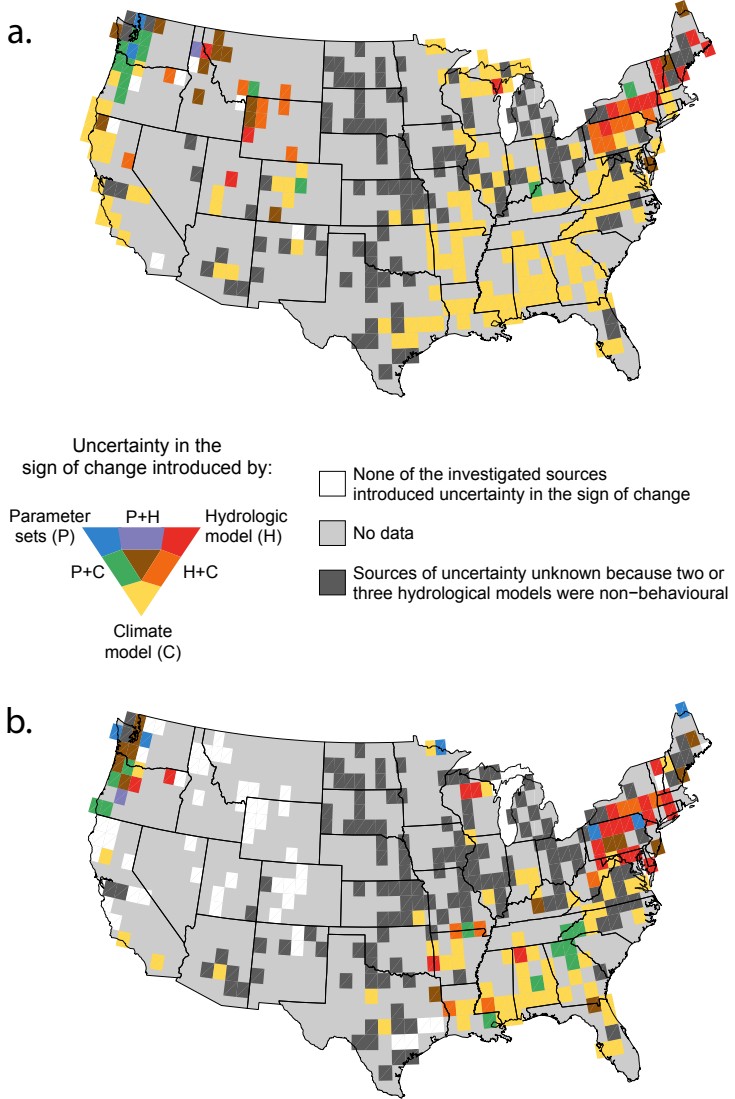

**Figure 9.** Distribution of the combined investigated sources of uncertainty for a) mean runoff, and b) discharge timing. Spatial coverage is obtained by determining a grid-based ($1 \times 1^{\circ}$) maximum likelihood, see Section 2.6. Results for different General Circulation Model (GCM) - hydrologic model combinations separately can be found in Appendix Figure A4 and B4. Note that three important sources of uncertainty have not been investigated in this study: internal climate variability, emission scenarios, and the statistical downscaling technique for GCM output.

east the climate models dominate as source of uncertainty, and in the north-east the hydrologic models lead to disagreement on the sign of change.

Comparing the two investigated metrics, Figure 9a and b, we recognize generally comparable spatial patterns in the sources of uncertainty, for example the role of GCMs in the south-east and hydrologic models in the north-east. There are, however, also some differences. In the north-west, the sign of change in discharge timing is influenced by all three investigated sources of uncertainty, whereas the sign of change in mean runoff in the north-west is mainly controlled by the climate model and the parameter sets. In the south-east, we see for both a region influenced by the GCM, but for the discharge timing we see spots in this region where hydrologic model and parameter sets are also important. The most remarkable difference between both metrics is that for the discharge timing, a large region in the mid-west and parts of the west coast can be identified where there is unanimous agreement on the sign of change (the white grids in Figure 9).

## 4  Discussion

We demonstrated that uncertainty in the projected changes in streamflow volume and timing are mainly controlled by the the GCM forcing, followed by the choice of the hydrologic model, and the parameter sets of the hydrologic models, respectively. In Figure 9, it is shown that parameter uncertainty particularly occurs in the north-west United States. These coastal basins are mainly precipitation-driven (Fritze et al., 2011) and receive, in the current climate, the highest precipitation sum in the US (see Figure 1 in Newman et al., 2015). Furthermore, these basins had the highest number of behavioural parameter sets for SAC and VIC (Appendix Figure C1). Several studies showed that humid circumstances enhance the calibration process (Melsen et al., 2014; Perrin et al., 2007; Yapo et al., 1996, e.g.), which can explain why so many parameters were classified as behavioural in these basins. This study, however, shows the downside of the 'easy' identification of parameters in humid catchments: Since so many different parameter sets are able to describe current discharge behaviour comparably well, it is difficult to distinguish which parameter sets capture the processes correctly, which increases the spread in future projections and subsequently leads to disagreement in the sign of change. As suggested in Section 3.1, one way of tackling this problem is to use observations from several states and fluxes to constrain the hydrologic models.

In the north-east United States, hydrologic model induced uncertainty is more prominent. In Section 3.2, this was related to snow processes. The three employed hydrologic models have different model conceptualizations for snow processes, which can explain the contrasting results: VIC solves the energy balance to determine snow accumulation and ablation, while SAC and VIC use only a temperature-driven approach. The exact mechanisms behind the impact on runoff of a precipitation shift from snow towards rain are not yet well understood (Berghuijs et al., 2014), but it should also be acknowledged that not all our knowledge on snow processes is incorporated in the employed hydrologic models. Several other studies also identified snow processes as critical in hydrologic projections (e.g. Dobler et al., 2012; Vidal et al., 2016). These results provide a strong motivation to carefully test the snow conceptualization in hydrologic models before applying them in climate change impact studies.

Especially in the south-east United States, disagreement in projections is mainly induced by the climate models, related to disagreement among the climate models on the change of precipitation (Section 3.3). An important conceptual assumption in

this study is the offline application of climate models and hydrologic models, i.e., there is no feedback between the hydrologic models and the climate models. The interaction of the climate with the land surface is now only represented through land surface models included in the GCM. Changing the parameters or conceptualization of the land surface (like we did in this study with the hydrologic models) would therefore influence the GCM projection. Milly and Dunne (2016) demonstrate that an offline application of GCMs to, for example, hydrologic models severely influences the potential evapotranspiration, mainly because (hydrologic) models do not account for changes in stomatal conductance. A fully coupled approach, although computationally expensive in an uncertainty analysis that requires many runs, would therefore give a more realistic overview of the spread in the projections, and could perhaps limit this spread.

Besides the spatial pattern recognized in the three investigated sources of uncertainty, three other important sources of uncertainty have not been considered in this study, of which one is the uncertainty in emission scenario, which depends on greenhouse gas emissions and the policy that will be implemented to limit emissions. Two other important sources of uncertainty are the statistical downscaling technique of the GCM output, and the internal climate variability. Gutmann et al. (2014) showed that different downscaling techniques each have their advantages and disadvantages. The technique employed here, Bias Corrected Spatial Disaggregation (Wood et al., 2004), tends to overestimate the wet-day fraction and underestimate extreme events, which both can influence the hydrologic response of the catchment. Accounting for the internal climate variability implies considering different realizations of the same GCM. It has been demonstrated that slightly different initial conditions can lead to a substantially different GCM realization (Deser et al., 2012; Fatichi et al., 2016). Furthermore, land-use and soil parameters have been kept constant in the hydrologic model for both modelling periods, although it is very likely that land use will change in the future. The climate projections account for land use change that is prescribed in the representative concentration pathway (in this study RCP8.5). These changes are, however, difficult to translate into the conceptual parameters of the employed hydrological models. Whereas VIC explicitly accounts for land use through e.g. root-zone thickness and stomatal resistance parameters, land use is not explicitly parameterized in SAC and HBV. Considering that at least three important sources of uncertainty have not been taken into account, this study likely presents an underestimation of the total uncertainty in the sign of change.

The substantial uncertainty in the sign of change, however, does not imply that it is impossible to say anything about future changes in our hydrologic system. We showed that the uncertainty in the sign of change is more wide-spread for the mean runoff metric than for the discharge timing metric. Addor et al. (2014) already showed that projected changes in the timing of discharge in Swiss catchments are significantly more robust (i.e. higher agreement among the ensemble members and greater deviation from the baseline) than changes in mean discharge. They also showed that the relative contribution of the different sources of uncertainty varies in space. Here, we use a greater number of catchments, covering a wider range of hydro-climatic conditions, to further explore what drives the uncertainty in hydrological projections. We found changes that are robust, and in regions where the changes are unclear, we determined what drives their uncertainty and outlined ways to reduce it.

# 5 Conclusions

The goal of this study was to reveal sources of uncertainty in hydrologic projections, and to provide directions for further research to decrease or account for this uncertainty, in order to define realistic quantitative hydrologic storylines. We focussed on the sign of change in mean annual runoff and discharge timing (day of the year where half of the discharge has passed). In our results, GCM forcing was the main source of uncertainty, followed by the hydrologic model structure and the parameters of the hydrologic model. Different sources of uncertainty dominated in different regions. In general, there was more agreement on the sign of change in discharge timing than on the sign of change in mean runoff. Three important sources of uncertainty have not been considered: emission scenario, GCM downscaling technique, and internal climate variability.

In some regions parameters and/or hydrologic model uncertainty were the most important source of uncertainty, illustrating the need for improved process representation and parameter estimation in hydrologic models. We could relate the uncertainty in the mean runoff metric to snow and aridity processes dictated by the hydrologic model, and uncertainty in the discharge timing metric to basins with a lower elevation (non-mountainous regions). Uncertainty in both metrics was influenced by (dis)agreement among GCMs regarding the projected change in precipitation. Furthermore, our study revealed a spatial pattern in the uncertainty of hydrologic projections: different locations need different priorities to reduce uncertainty.

Our study included many of the headwaters of reservoirs that are currently used to store water for domestic supplies and irrigation, for example the Delaware River that feeds the Pepacton reservoir, the main drinking water reservoir for New York City. The compelling social relevance of these basins provides a strong motivation to account for uncertainty in water management decisions, by using our understanding of uncertainty to develop quantitative hydrologic storylines.

*Code and data availability.* All codes to process the model results (mainly Matlab) and the model results itself are available upon request by the corresponding author. Forcing and observed runoff for the complete dataset of catchments is publicly available, see Newman et al. (2014, 2015). Catchment characteristics are publicly available in the CAMELS dataset (Addor et al., 2017a, b).

## Appendix A: Overview of all model combinations for mean runoff metric

## Appendix B: Overview of all model combinations for half-discharge metric

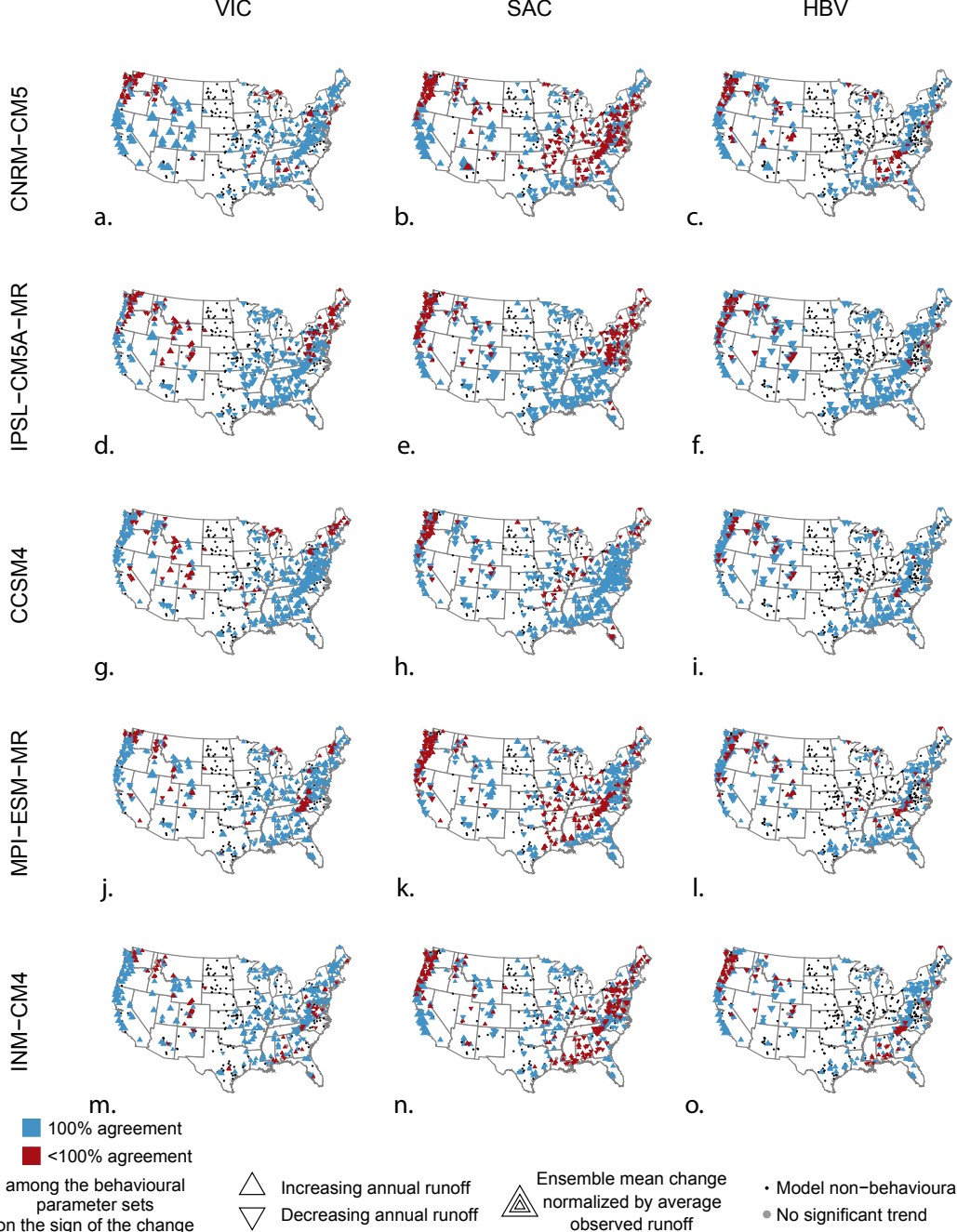

**Figure A1.** The agreement among the different model runs (representing different behavioural parameter sets) on the sign of the ensemble mean change in mean annual runoff for three hydrologic models (columns, VIC = Variable Infiltration Capacity Model, SAC = Sacramento Soil Moisture Accounting Model, HBV = Hydrologiska Byråns Vattenbalansavdelning Model) forced with five different climate models (rows). The direction of the triangle-marker shows the sign of the ensemble mean change, the size of the marker indicates the relative projected change. a) VIC forced with CNRM-CM5. b) SAC forced with CNRM-CM5. c) HBV forced with CNRM-CM5. d) VIC forced with IPSL-CM5A-MR. e) SAC forced with IPSL-CM5A-MR. f) HBV forced with IPSL-CM5A-MR. g) VIC forced with CCSM4. h) SAC forced with CCSM4. i) HBV forced with CCSM4. j) VIC forced with MPI-ESM-MR. k) SAC forced with MPI-ESM-MR. l) HBV forced with MPI-ESM-MR. m) VIC forced with INM-CM4. n) SAC forced with INM-CM4. o) HBV forced with INM-CM4.

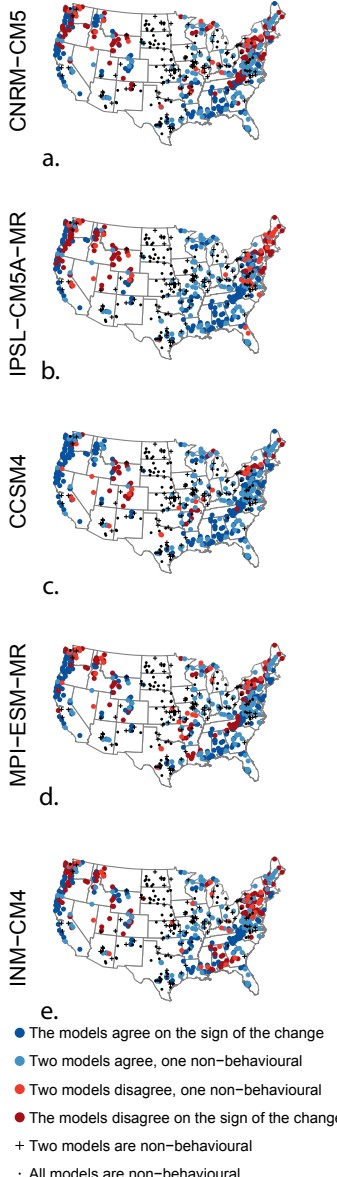

**Figure A2.** The agreement among the three hydrologic models on the sign of the ensemble mean change in mean annual runoff, forced with five different climate models. a) The three hydrologic models are forced with CNRM-CM5 data. b) The three hydrologic models are forced with IPSL-CM5A-MR data. c) The three hydrologic models are forced with CCSM4 data. d) The three hydrologic models are forced with MPI-ESM-MR data. e) The three hydrologic models are forced with INM-CM4 data.

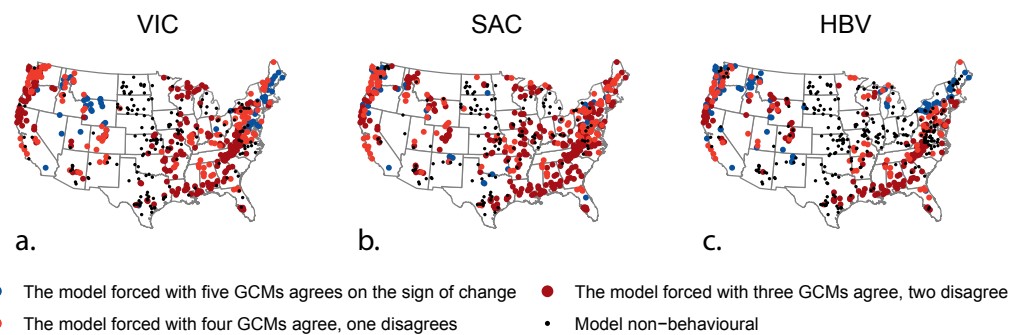

**Figure A3.** The agreement on the sign of the ensemble mean change in mean annual runoff in the output from the same hydrologic model forced with five different climate models. a) Agreement when the Variable Infiltration Capacity Model (VIC) is forced with data from five different GCMs. b) Agreement when Sacramento Soil Moisture Accounting Model (SAC) is forced with data from five different climate models. c) Agreement when Hydrologiska Byråns Vattenbalansavdelning Model (HBV) is forced with data from five different GCMs.

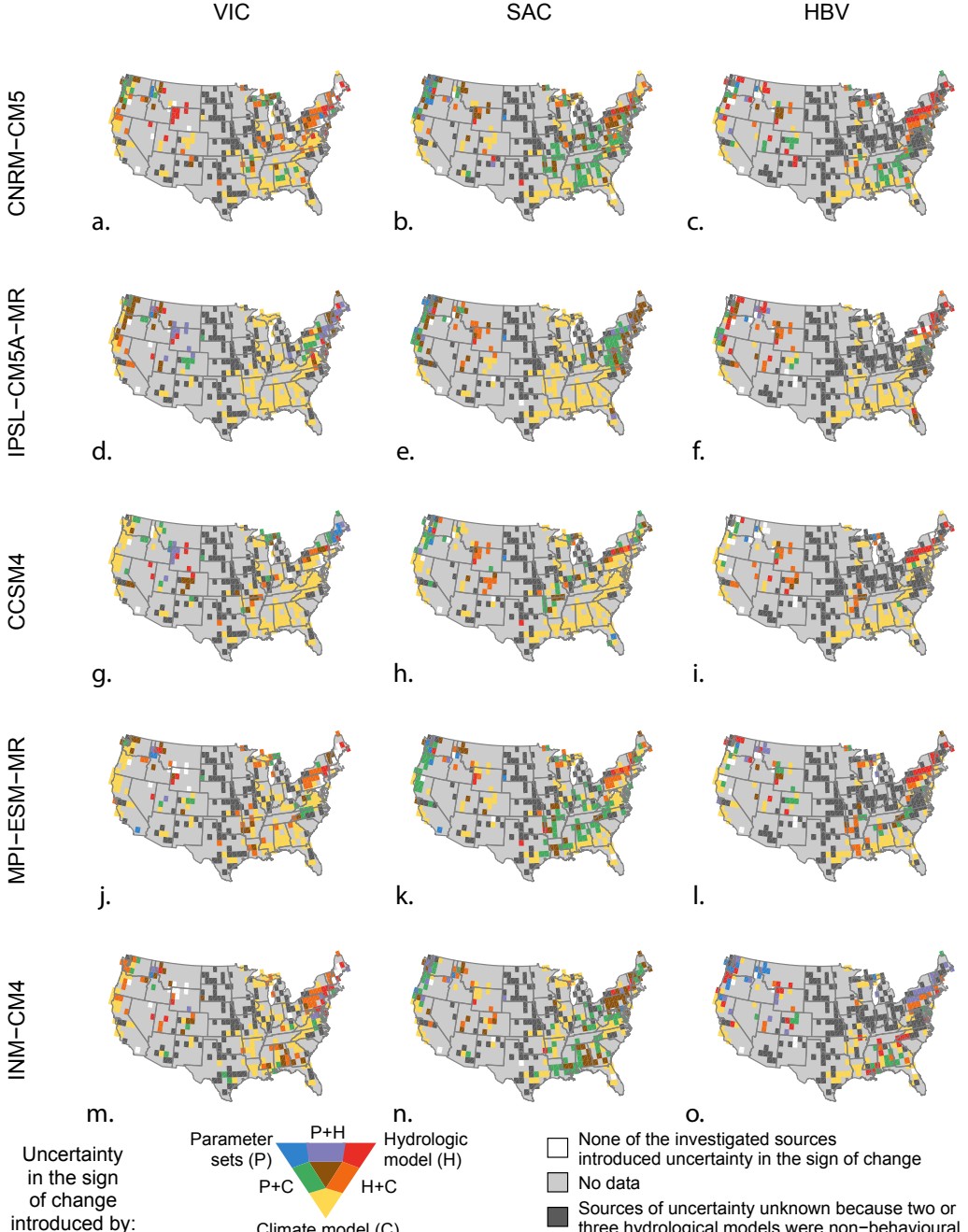

**Figure A4.** Distribution of the combined investigated sources of uncertainty for mean annual runoff. Spatial coverage is obtained by determining a grid-based maximum likelihood. a) Variable Infiltration Capacity Model (VIC) was used as reference when climate model uncertainty was tested, CNRM-CM5 was used as reference when the hydrologic models were tested. b) Sacramento Soil Moisture Accounting Model (SAC) and CNRM-CM5 as reference options. c) Hydrologiska Byråns Vattenbalansavdelning Model (HBV) and CNRM-CM5 as reference options. d) VIC and IPSL-CM5A-MR as reference options. e) SAC and IPSL-CM5A-MR as reference options. f) HBV and IPSL-CM5A-MR as reference options. g) VIC and CCSM4 as reference options. h) SAC and CCSM4 as reference options. i) HBV and CCSM4 as reference options. j) VIC and MPI-ESM-MR as reference options. k) SAC and MPI-ESM-MR as reference options. l) HBV and MPI-ESM-MR as reference options. m) VIC and INM-CM4 as reference options. n) SAC and INM-CM4 as reference options. o) HBV and INM-CM4 as

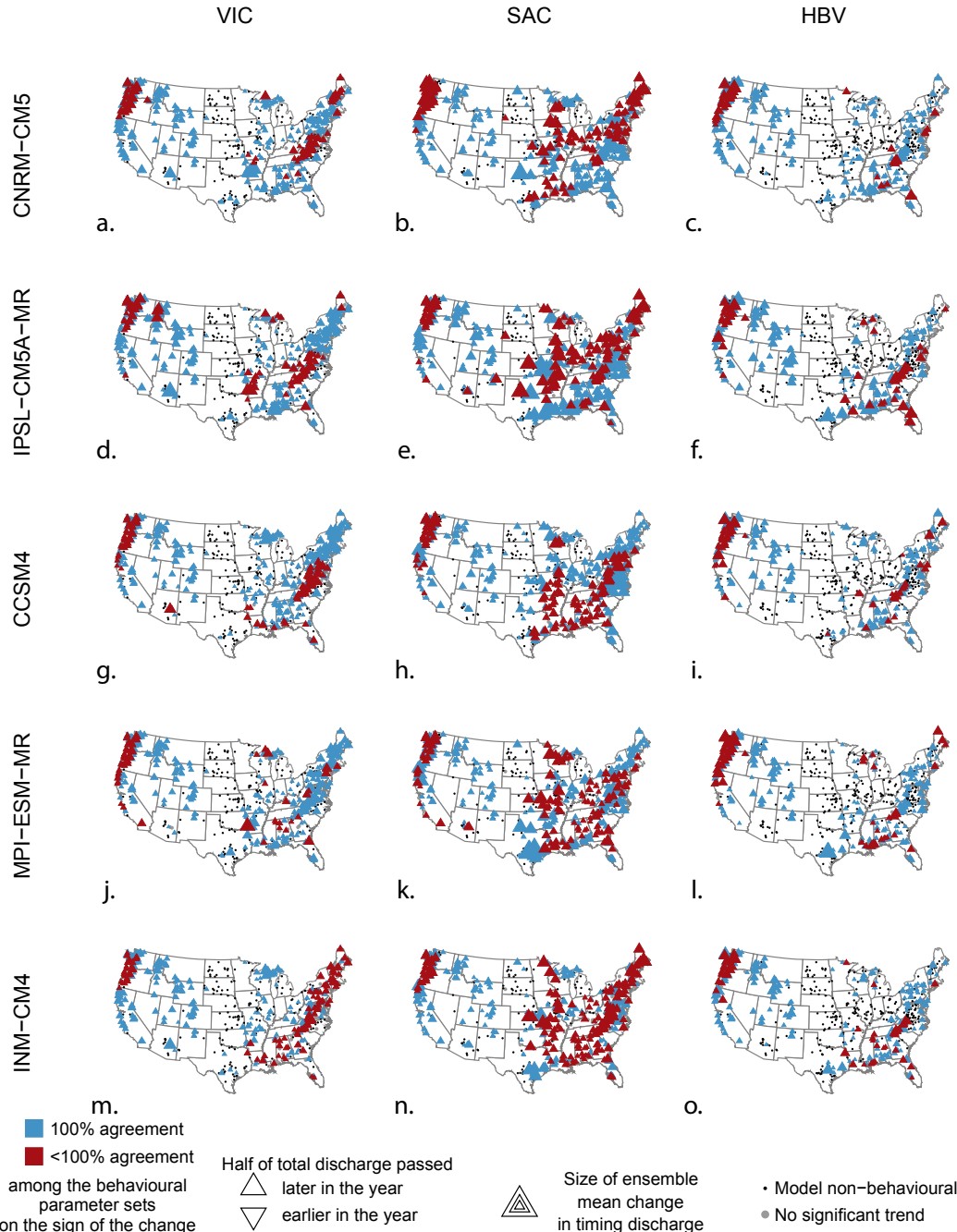

**Figure B1.** The agreement among the different model runs (representing different behavioural parameter sets) on the sign of change in discharge timing for three hydrologic models (columns, VIC = Variable Infiltration Capacity Model, SAC = Sacramento Soil Moisture Accounting Model, HBV = Hydrologiska Byråns Vattenbalansavdelning Model) forced with five different General Circulation Models (GCMs, rows). The direction of the triangle-marker shows the sign of the ensemble mean change, the size of the marker indicates the relative projected change. a) VIC forced with CNRM-CM5. b) SAC forced with CNRM-CM5. c) HBV forced with CNRM-CM5. d) VIC forced with IPSL-CM5A-MR. e) SAC forced with IPSL-CM5A-MR. f) HBV forced with IPSL-CM5A-MR. g) VIC forced with CCSM4. h) SAC forced with CCSM4. i) HBV forced with CCSM4. j) VIC forced with MPI-ESM-MR. k) SAC forced with MPI-ESM-MR. l) HBV forced with MPI-ESM-MR. m) VIC forced with INM-CM4. n) SAC forced with INM-CM4. o) HBV forced with INM-CM4.

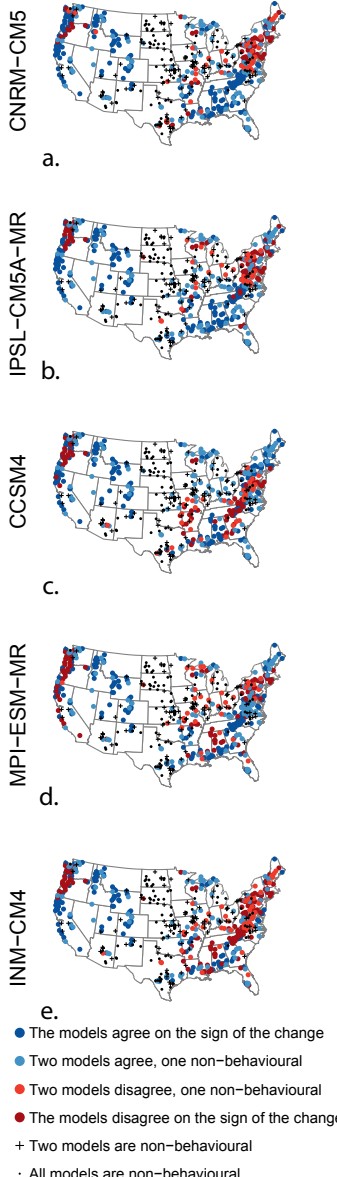

**Figure B2.** The agreement among the three hydrologic models on the sign of the ensemble mean change in discharge timing, forced with five different climate models. a) The three hydrologic models are forced with CNRM-CM5 data. b) The three hydrologic models are forced with IPSL-CM5A-MR data. c) The three hydrologic models are forced with CCSM4 data. d) The three hydrologic models are forced with MPI-ESM-MR data. e) The three hydrologic models are forced with INM-CM4 data.

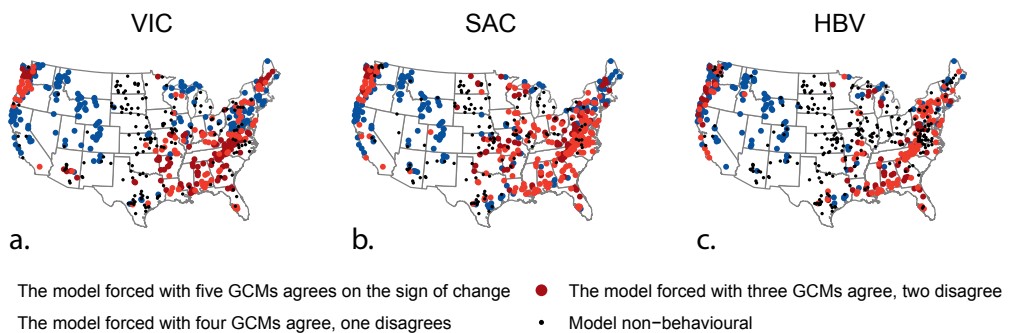

**Figure B3.** The agreement on the sign of the ensemble mean change in discharge timing in the output from the same hydrologic model forced with five different climate models. a) Agreement when Variable Infiltration Capacity Model (VIC) is forced with data from five different GCMs. b) Agreement when Sacramento Soil Moisture Accounting Model (SAC) is forced with data from five different climate models. c) Agreement when Hydrologiska Byråns Vattenbalansavdelning Model (HBV) is forced with data from five different GCMs.

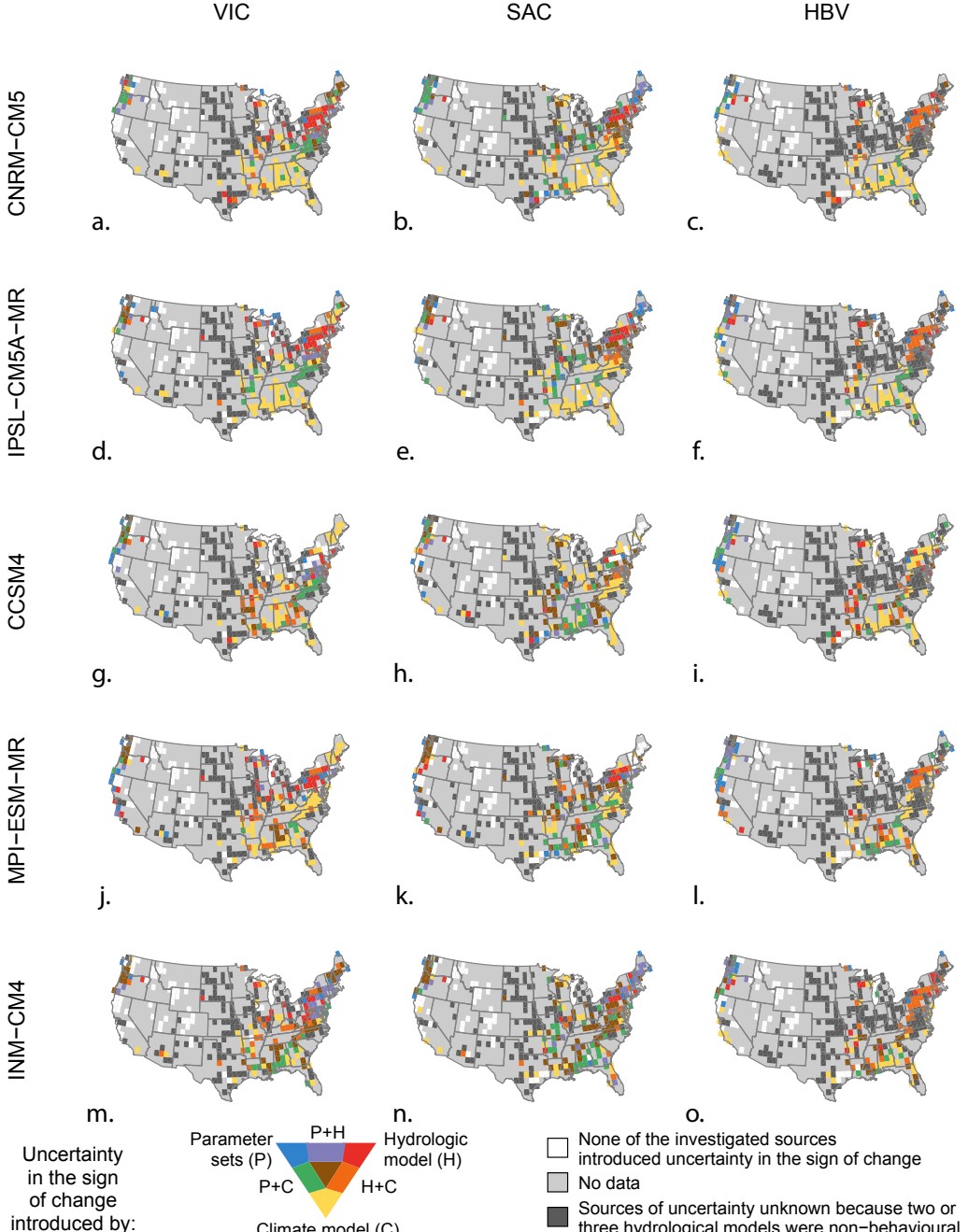

**Figure B4.** Distribution of the combined investigated sources of uncertainty for the discharge timing. Spatial coverage is obtained by determining a grid-based maximum likelihood. a) Variable Infiltration Capacity Model (VIC) was used as reference when climate model uncertainty was tested, CNRM-CM5 was used as reference when the hydrologic models were tested. b) Sacramento Soil Moisture Accounting Model (SAC) and CNRM-CM5 as reference options. c) Hydrologiska Byråns Vattenbalansavdelning Model (HBV) and CNRM-CM5 as reference options. d) VIC and IPSL-CM5A-MR as reference options. e) SAC and IPSL-CM5A-MR as reference options. f) HBV and IPSL-CM5A-MR as reference options. g) VIC and CCSM4 as reference options. h) SAC and CCSM4 as reference options. i) HBV and CCSM4 as reference options. j) VIC and MPI-ESM-MR as reference options. k) SAC and MPI-ESM-MR as reference options. l) HBV and MPI-ESM-MR as reference options. m) VIC and INM-CM4 as reference options. n) SAC and INM-CM4 as reference options. o) HBV and

## Appendix C: Overview of sampled parameters in the hydrological models

**Table C1.** Selected parameters and their boundaries for the Variable Infiltration Capacity Model (VIC) model (see Methods). Parameter 1 to 7 were shown to be the most sensitive parameters based on Demaria et al. (2007); Chaney et al. (2015) and Melsen et al. (2016). Parameter 8 to 14 were selected for their impact on snow and/or evapotranspiration processes. Parameter 15 - 17 are usually hard-coded in the VIC model, but were shown to be highly sensitive in Mendoza et al. (2015b) and are therefore included in the sampling. LB = lower boundary, UB = upper boundary.

|  | Name | Unit | LB | UB | Description |
|---|---|---|---|---|---|
| 1 | $B_i$ | - | $10^{-5}$ | 0.4 | Infiltration shape parameter |
| 2 | $D_s$ | - | $10^{-4}$ | 1.0 | Fraction of $D_{s,max}$ where non-linear baseflow starts |
| 3 | $D_{s,max}$ | $mm\ d^{-1}$ | 0.1 | 50 | Max velocity of the baseflow |
| 4 | $W_s$ | - | 0.2 | 1.0 | Fraction of $W_{s,max}$ where non-linear baseflow starts |
| 5 | $Expt_2$ | - | 4.0 | 30 | Exponent of the Brooks-Corey relation |
| 6 | $Depth_2$ | m | 0.1 | 3.0 | Depth of soil layer 2 |
| 7 | $Depth_3$ | m | 0.1 | 3.0 | Depth of soil layer 3 |
| 8 | $Ts_{max}$ | °C | 0.0 | 3.0 | Max temperature where snowfall can occur |
| 9 | $Ts_{min}$ | °C | $Ts_{max}$-0.01 | $Ts_{max}$-3.0 | Min temperature where rainfall can occur |
| 10 | SR | - | $5^{-5}$ | 0.5 | Surface roughness of the snow pack |
| 11 | $RZT_1$ | - | 0.5 | 2 | Multiplication factor for rootzone thickness layer 1 |
| 12 | $RZT_2$ | - | 0.5 | 2 | Multiplication factor for rootzone thickness layer 2 |
| 13 | $RZT_3$ | - | 0.5 | 2 | Multiplication factor for rootzone thickness layer 3 |
| 14 | $R_{min}$ | - | 0.1 | 10 | Multiplication factor for minimum stomatal resistance of the vegetation |
| 15 | newalb | - | 0.7 | 0.99 | New snow albedo |
| 16 | albaa | - | 0.88 | 0.99 | Base in snow albedo function for accumulation |
| 17 | albtha | - | 0.66 | 0.98 | Base in snow albedo function for melt |

**Table C2.** Selected parameters and their boundaries for the Sacramento Soil Moisture Accounting Model (SAC) model (see Methods). The parameter boundaries are based on Newman et al. (2015), the Priestley-Taylor parameter (number 18) has been adapted based on Lhomme (1997). LB = lower boundary, UB = upper boundary.

| | Name | Unit | LB | UB | Description |
|---|---|---|---|---|---|
| 1 | MFAX | mm $°C^{-1}$ $6h^{-1}$ | 0.8 | 3.0 | Max melt factor |
| 2 | MFMIN | mm $°C^{-1}$ $6h^{-1}$ | 0.01 | 0.79 | Min melt factor |
| 3 | UADJ | km $6h^{-1}$ | 0.01 | 0.40 | Wind adjustment factor for rain on snow |
| 4 | SI | mm | 1.0 | 3500 | snow water equivalent for 100% snow area |
| 5 | SCF | - | 0.1 | 5.0 | Undercatch correction factor |
| 6 | PXTEMP | °C | -3.0 | 3.0 | Temperature for rain/snow transition |
| 7 | UZTWM | mm | 1.0 | 800 | Upper zone max storage of tension water |
| 8 | UZFWM | mm | 1.0 | 800 | Upper zone max storage of free water |
| 9 | LZTWM | mm | 1.0 | 800 | Lower zone max storage of tension water |
| 10 | LZFPM | mm | 1.0 | 800 | Lower zone max storage of free water |
| 11 | LZFSM | mm | 1.0 | 1000 | Lower zone max storage of secondary free water |
| 12 | UZK | $day^{-1}$ | 0.1 | 0.7 | Upper zone free water lateral depletion rate |
| 13 | LZPK | $day^{-1}$ | $1^{-5}$ | 0.025 | Lower zone primary free water depletion rate |
| 14 | LZSK | $day^{-1}$ | $1^{-3}$ | 0.25 | Lower zone secondary free water depletion rate |
| 15 | ZPERC | - | 1.0 | 250 | Max percolation rate |
| 16 | REXP | - | 0.0 | 6.0 | Exponent of the percolation equation |
| 17 | PFREE | - | 0.0 | 1.0 | Fraction percolating from the upper to the lower zone |
| 18 | P-T | - | 1.0 | 1.74 | Priestley-Taylor coefficient |

**Table C3.** Selected parameters and their boundaries for the Hydrologiska Byråns Vattenbalansavdelning Model model (see Methods). The selected parameters are based on Parajka et al. (2007), the parameter boundaries have been widened based on Uhlenbrook et al. (1999) and Abebe et al. (2010). The Priestley-Taylor parameter (number 15) is based on Lhomme (1997). LB = lower boundary, UB = upper boundary.

| | Name | Unit | LB | UB | Description |
|---|---|---|---|---|---|
| 1 | SCF | - | 0.1 | 5.0 | Snow correction factor |
| 2 | DDF | mm $^{\circ}$C$^{-1}$ day$^{-1}$ | 0.04 | 12 | Degree day factor |
| 3 | Tr | $^{\circ}$C | 0.0 | 3.0 | Temperature above which precipitation is rain |
| 4 | Ts | $^{\circ}$C | Tr-0.01 | Tr-3 | Temperature below which precipitation is snow |
| 5 | Tm | $^{\circ}$C | -3.0 | 3.0 | Temperature where melt starts |
| 6 | LP | - | 0.0 | 1.0 | Evaporation reduction threshold |
| 7 | FC | mm | 0.0 | 2000 | Max soil moisture storage |
| 8 | BETA | - | 0.0 | 20 | Non-linear shape coefficient |
| 9 | K0 | day | 0.0 | 2.0 | Storage coefficient of very fast response |
| 10 | K1 | day | 2.0 | 30 | Storage coefficient of fast response |
| 11 | K2 | day | 30 | 250 | Storage coefficient of slow response |
| 12 | L | mm | 0.0 | 100 | Reservoir threshold |
| 13 | PERC | mm day$^{-1}$ | 0.0 | 100 | Percolation rate |
| 14 | BMAX | day | 0.0 | 30 | Max baseflow of low flows |
| 15 | P-T | - | 1.0 | 1.74 | Priestley-Taylor coefficient |

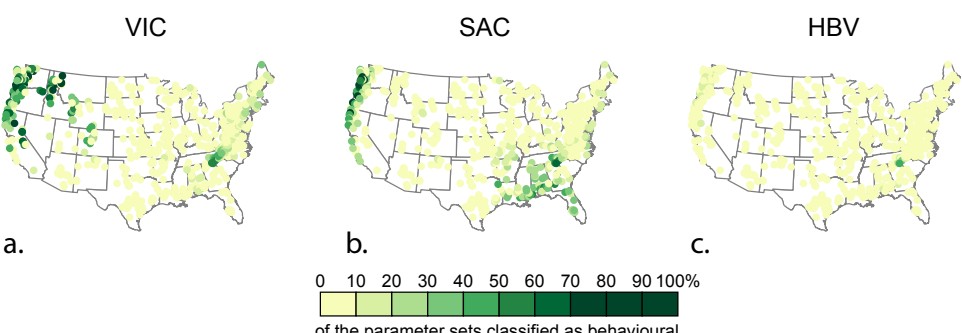

**Figure C1.** The percentage of parameter sets that have been classified as behavioural based on the Kling-Gupta criterion. a) for the Variable Infiltration Capacity Model (VIC), for a total parameter sample of 1800. b) for the Sacramento Soil Moisture Accounting Model (SAC), with a total parameter sample of 1900. c) for the Hydrologiska Byråns Vattenbalansavdelning Model (HBV) with a total parameter sample of 1600.

*Author contributions.* LM and AT designed the study in consultation with MC, NA and RU. LM carried out the analyses. NM and AN constructed the SAC and VIC model set-up and prepared the forcing files. NA and AN prepared the catchment characteristics files. PT suggested the statistical tests. All authors contributed to the interpretation and reporting of the results.

*Competing interests.* There are no competing interests.

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
