# Peer review of "Mapping (dis)agreement in hydrologic projections"

_Hydrology and Earth System Sciences, 2017_

## Referee Comment (RC1) · J. Kiesel (Referee) · 11 Nov 2017

The manuscript "Mapping (dis)agreement in hydrologic projections" by Melsen et al. shows an extensive analysis of selected uncertainties inherited in climate change projections. The authors consider a cascade of uncertainties: three different hydrologic models; different model parameterizations that pass a performance threshold; five different climate models; 605 catchments across the contiguous US. They analyse these combinations on their direction of change of future mean discharge and timing of discharge. The results show significant disagreement in the direction of change for each uncertainty component.

The paper is well structured, well written, the results are well presented and the discussion and conclusion cover the major points. I therefore suggest minor revision.

The major comment I have is related to a more detailed discussion of the behavioural

model runs. Please find detailed comments regarding this and other issues below.

METHODOLOGY

p.2 l.26: I suggest to mention the information about which RCP you used in chapter 2.2

p.3 Fig 1d is not clear to me: why two bars in the upper positive change section? What is the black line with the two points and how is it created? Why is 'Frequency' written below the box?

p.4 l.1ff: Model description: I miss one sentence for explaining the model concept about runoff components (surface, lateral, groundwater) which are important for discharge timing

p.4 l.3: moisture

p.5 l.1: what are the "100 base runs with average parameter values" - how were these defined? e.g. nothing about this is mentioned in the parameter tables (Annexe)

p.5 l.4: I miss one or two sentences about the other input data. I know it is publicly available in the CAMELS dataset. But I think it is important to know the very basics here: Topography, land use, soil/geology and if catchment management (irrigation, damming) is considered and included in the dataset and in models. If catchment management is not included, could that be a reason for the non-behavioural catchments? I.e. the central US is subject to the highest ratio of agriculture and this distribution seems to fit well to the non-behavioural spots (https://www.agcensus.usda.gov/Publications/2007/Online_Highlights/Ag_Atlas_Maps/Farms/Land_in_Farms_and_Land_M085.php). If relevant, this could be an additional point for the discussion (see my comment at p.8 l.18ff).

p.5 l.8: For future studies, consider using the KGE': "For the variability ratio c we used CVs/CVo instead of rs/ro, which was proposed in the original version of the KGE-statistic (Gupta et al., 2009). This ensures that the bias and variability ratios are not cross-correlated, which otherwise may occur when e.g. the precipitation inputs are

biased." Kling et al. 2012, JOH.

p.5 l.16 and p.6 l.2: Please comment on why the two time periods differ in length. When comparing aggregated/average metrics the length of the comparison period is important since the longer the time series, the less influenced are the metrics by singular annual extremes.

p.5 l.18: I suggest to explain the choice of the rcp briefly in a sentence.

p.5 l.20: what is meant by "member" (i.e. what distinguishes the different 'members' of each GCM family - regional climate model, version, resolution, year, ....)?

p.6 l.6-7: I suggest to define what you mean by "ensemble mean change". E.g. I think something along these lines is clearer: "The ensemble mean change was then determined as the mean change over all behavioural parameter sets of each GCM-, hydrological model-, and catchment combination".

p.6 l.12: I assume number of "representative sample of parameter sets" is defined through the behavioural runs. For the other two uncertainty sources we know the number (three for the hydro models, five for the GCMs) - but for the chosen parameter sets you do not show them. However, I think it matters how many runs in each catchment are used to produce all the subsequent results. Could you show three additional maps of the CONUS (could also go to the Appendix) where the color of each catchment dot indicates the number of behavioural runs for each hydrologic model?

p.6 l.18-26: I really like this part of the analysis, but the paragraph is difficult to understand without having seen the results and I suggest to begin the paragraph with an explanation, e.g. something along the lines: "It is assumed that catchment characteristics can influence the agreement between hydrological models and GCM. To assess the influence on the hydrological model agreement, we divided all basins into three categories: ... "

RESULTS

p.8 l.7-8: Is this result not better suited for the section 3.2.1?

p.8 l.18ff: I think this is an important paragraph which is valid for the other sources of uncertainty as well. I.e. if you end up with model runs that generally depict the processes better, you may end up with less disagreement for the other uncertainty causes as well. So, an interesting hypothesis to test would be, if most of your mapped disagreement is caused by parameter sets at the lower end of the KGE and if high KGEs lead to higher agreement (though I am interested in this, this is just a side note, no need to do this within this paper).

However, I think the paragraph fits better to the discussion and I suggest to add:

- that it could be possible that improved process depiction in your models could reduce disagreement related to other uncertainty sources as well

- numerous studies (e.g. Pool et al. 2017 HESS 21) have found that looking at certain metrics without having used them in the optimization (in your case: selection of behavioural runs) can cause inadequate depiction of those metrics - so the actual selection of the objective function may influence (dis-)agreement

- a short statement if you can rule out that the non-behavioural results could be due to the selection of the parameters and ranges which may be more suitable for conditions significantly different from the catchments that are non-behavioural

p.8 l.29: Suggest to change to "... hydrologic models (dis)agree on the sign of..."

p.8 l.30: "in the north-east the models..."

p.11 l.6: suggest to replace "...was able to capture current..." through "...was non-behavioural (Figure 4e)"

p.11. l.29: a lower aridity? Figure 5.d. suggests higher aridity? how can aridity be both high for disagreement and non-behavioural catchments? also at Figure 7.d: the significance triangle for mean delta P should point down and be hollow or? Seems
like I have difficulties understanding the rose plots. If the plots are correct, I require more explanation how they need to be interpreted (e.g. already in the methods with an example rose plot).

p.15 l.10: This chapter is a very good summary of the uncertainties. But I miss that you explain how the combined uncertainty is produced in the methods. Did I miss something?

DISCUSSION

p.18 l.1: Ehret et al. 2012 (HESS Opinions) is also a suitable reference to this statement.

CONCLUSION

I suggest to add a few sentences which hydrologic model, which GCM and which combination led to the highest (dis)agreement. I know that the information is scattered throughout the results, but I would have liked to see this information summarized in the conclusions. I envision these sentences as a very concise summary of the whole Appendix.

---

## Author Comment (AC1) · 15 Nov 2017

Dear Dr Kiesel,

Thank you very much for your positive and constructive review. Please find below a short response to a selection of the points raised by you. We only respond to the points that caused confusion in understanding the manuscript, please let us know if anything remains unclear or if you disagree. We plan to incorporate all your suggestions if we get the opportunity to revise our manuscript.

Best regards,

on behalf of all co-authors, Lieke Melsen

*p.3 Fig 1d is not clear to me: why two bars in the upper positive change section? What is the black line with the two points and how is it created? Why is 'Frequency' written

below the box?

The changes from the six points compared to the 1:1 line in Figure 1c are summarized in a histogram as shown in figure 1d. This histogram is turned 90degrees, and therefore frequency is shown on the x-axis. The black line shows the mean change from the distance from the six points in 1c to the 1:1 line. We will adapt the caption to make this more clear.

*p.5 l.20: what is meant by "member" (i.e. what distinguishes the different 'members' of each GCM family - regional climate model, version, resolution, year, ....)?

Knutti et al (2013) defined GCM families based on their output: "based on the predicted change in temperature and precipitation fields for the end of the 21st century in the RCP8.5 scenario relative to the control." (Figure 1 in Knutti et al., 2013). By selecting one member (GCM) of each GCM-family, we approach the full range of projections by all GCMs. We will add this to the text.

*p.6 l.12: I assume number of "representative sample of parameter sets" is defined through the behavioural runs. For the other two uncertainty sources we know the number (three for the hydro models, five for the GCMs) - but for the chosen parameter sets you do not show them. However, I think it matters how many runs in each catchment are used to produce all the subsequent results. Could you show three additional maps of the CONUS (could also go to the Appendix) where the color of each catchment dot indicates the number of behavioural runs for each hydrologic model?

Yes, we can definitely do that, thank you for the suggestion. It actually also provides relevant information; generally speaking, the regions where there is disagreement based on parameters, are the regions where a large number of parameter-sets was considered behavioural (probably, because these regions are generally speaking quite wet, which is, again generally speaking, easier to model).

*p.8 l.7-8: Is this result not better suited for the section 3.2.1?

We think not, because these results specifically refers to parameter-disagreement, for a specific model. We admit, however, that this can be confusing, and will add a sentence to explain. It is, indeed, not always a clear distinction between what is part of parameter uncertainty and what of model structure, as parameters are representatives of the model structure.

*p.11. l.29: a lower aridity? Figure 5.d. suggests higher aridity? how can aridity be both high for disagreement and non-behavioural catchments?

Thank you, it should indeed be higher aridity. The aridity can be higher for both the disagreement and non-behavioural catchments because for the analysis in the rose-plots, each group is compared to the total of the other groups (in other words; the disagreement-group is compared to all other groups, so the agreement and the non-behavioural group together). This indicates that the aridity in the agreement-group is so much lower, that it results in a significantly higher aridity for both other groups.

*also at Figure 7.d: the significance triangle for mean delta P should point down and be hollow or? Seems like I have difficulties understanding the rose plots. If the plots are correct, I require more explanation how they need to be interpreted (e.g. already in the methods with an example rose plot).

Thank you, we understand the confusion here, mean delta P should indeed be a down-ward hollow triangle. I think this is a remnant from an earlier analysis, where we also still accounted for the non-behavioural basins in this analysis; these basins on average experience a lower change in delta P. We will adapt the figure. The other results in the figure are correct.

*p.15 l.10: This chapter is a very good summary of the uncertainties. But I miss that you explain how the combined uncertainty is produced in the methods. Did I miss something?

There was not really any more methodology involved rather than what is explained

in the caption of Figure 9; For figure 9, we combined all the previous figures in the manuscript (plus all the figures in the appendix to account for the different models) and determined the most frequent sources-of-uncertainty-combination. We will add this explanation to the methodology.

---

## Referee Comment (RC2) · Anonymous Referee #2 · 16 Nov 2017

The paper titled "Mapping (dis)agreement in hydrology projections" is a very interesting study that aims to quantify the different sources of uncertainty in future hydrologic projections; these sources of uncertainty include climate model uncertainty, model structure uncertainty, and model parameter uncertainty. The authors show using 605 basins over the contiguous United States how future changes in annual mean runoff and discharge timing can be impacted by these three different sources of uncertainty. The paper is well structured and well written and should be published in HESS. However, I have a series of comments and questions that will need to be addressed prior to publication.

Main comments:

- I was very impressed with the depth of the introduction, methods, and results. However, I was disappointed by the discussion section. I believe that there are many things

that could (and should!) be discussed regarding the implications of this work that would be a missed opportunity to not include in the discussion section. For example, what is the physical explanation (if any) of why you get the results that you do? There is plenty of material in the results to enable this discussion and I believe it would be quite useful.

- Another topic of discussion is also another main comment. One of the underlying assumptions that is made in this and all offline studies that use GCM output as input to an offline hydrologic model is that the differences in hydrologic projections based on using different model parameters and hydrologic models does not impact the climate system (i.e., feedbacks). In other words, the precipitation and temperature that you are using from the GCM output depend on a land surface model that is contained within its original GCM; changing the model and parameters will directly change the temperature and precipitation you get... As a result, the most complete way to approach this type of study would be to use your approach in coupled GCMs. However, I completely understand why this is computationally not feasible when you are looking at all the different sources of uncertainty. That being said, I believe it is important to at least discuss this problem and mention how you believe the results might change if it were feasible to do this study "online" instead of "offline". See the following paper for more background on this issue: "Milly, P. C., & Dunne, K. A. (2016). Potential evapotranspiration and continental drying. Nature Climate Change, 6(10), 946-949"

- Was the GCM output downscaled and bias corrected against the forcing used in the 1980-2008 simulations? I understand that it has indeed been bias corrected; however, bias correction is always done against some reference database; I would hope that that reference database is the one being used to force the 1980-2008 simulations. Please clarify. If it was not bias corrected against the forcings used for the 1980-2008 simulations, I am concerned that some of your signal in change in annual mean runoff could be attributed simply to discrepancies between the observed forcing and the "bias-corrected" GCM output.

- Using 5 years for spin-up is awfully low. Are you sure that this is appropriate? I would
have suggested cycling through the 1980 to 2008 a few times. Although this most likely does not disqualify the results, there should be some argument in the paper for why only 5 years appeared to be enough.

Other comments:

p4,l3: available moist?

- Section 3.1.2: "A larger parameter sample could therefore decrease the number of non-behavioural basins and even allow for a more stringent selection criterion." Maybe, but certainly not necessarily. Assuming that the LHS sample is robust, then your model parameters will already capture the details of the parameter space. I agree that given your relatively small sample size, there might be regions of the parameter space that you are disregarding. However, I suspect that it will be fairly minimal.

- Section 4: "Climate models disagree more in a more extreme scenario" - How much of this can be attributed simply to the disconnect between your models and the climate models themselves? (See issue regarding offline/online models above)

- "Furthermore, land-use and soil parameters have been kept constant for both modelling periods. Although it is very likely that land-use will change in the future as a result of climate change or population growth, there are currently no methods to quantify this change and translate that to parameter values for the employed hydrologic models." -> GCMs actually do currently account for land use change. Now to be fair, these changes are mostly due to offline studies that predict that land use will be in the future, however, this information is out there and could be used in theory within the hydrologic models.

---

## Author Comment (AC2) · 28 Nov 2017

Dear reviewer,

Thank you very much for your constructive suggestions. Below you find a response to the points made.

*Discussion

We agree that the topics mentioned by the reviewer are currently lacking in the discussion, but definitely deserve a place there. Physical explanation can be given in some instances, for example for the result that snow is one of the causes for disagreement, because the three different hydrologic models use different snow parameterizations (degree-day versus energy-balance), the problem with aridity and the related soil moisture-evaporation feedbacks could perhaps be linked to the off-line application of

the GCMs which the reviewer justly refers to. We will investigate more physical expla-
nations and include a discussion on the offline/online application of GCMs to hydrologic
models.

*Was the GCM output downscaled and bias corrected against the forcing used in the
1980-2008 simulations? I understand that it has indeed been bias corrected; however,
bias correction is always done against some reference database; I would hope that that
reference database is the one being used to force the 1980-2008 simulations. Please
clarify.

The GCM output was bias corrected on the Maurer-dataset for the period 1950-1999,
so this is a different dataset than where the 1980-2008 simulations have been per-
formed with (which was the Daymet-dataset). We compared the temperature and pre-
cipitation differences between Daymet and Maurer for the 605 basins. Generally, both
data-sets agree pretty well (similar in the mean), although precipitation differences exist
in the west, and there are temperature differences in the central US and in the higher
basins.

We expect that the effect of the difference in calibration-dataset and bias-correction-
dataset on the conclusion of our study is limited, since we do not compare the Daymet-
forced model results with the GCM forced model results, but the two GCM-forced model
results (historical versus future).

So the effect of this bias correction only applies to the calibration. There is certainly
the possibility that the calibration will be different using the Maurer-dataset rather than
the Daymet-dataset. We expect that the effect is limited because we use a parameter-
sample, but to test this, we decided that we will force a small number of basins with the
Maurer-dataset and investigate the effect on the calibration.

*Using 5 years for spin-up is awfully low. Are you sure that this is appropriate? I would
have suggested cycling through the 1980 to 2008 a few times. Although this most likely
does not disqualify the results, there should be some argument in the paper for why

only 5 years appeared to be enough.

For the most complex hydrologic model included in this study, VIC, when run on an hourly basis, three months was shown to be sufficient (Melsen et al., 2016, https://doi.org/10.5194/hess-20-2207-2016). For HBV, it has been stated that one year warm-up period on a daily basis is sufficient in most cases (Seibert and Vis, 2012; www.hydrol-earth-syst-sci.net/16/3315/2012/). Therefore we assumed five years to be ample. An important critical note here is indeed that the warm-up period can be longer in drier climates, which could perhaps explain the lower model performance in these regions.

*Section 3.1.2: "A larger parameter sample could therefore decrease the number of non-behavioural basins and even allow for a more stringent selection criterion." Maybe, but certainly not necessarily. Assuming that the LHS sample is robust, then your model parameters will already capture the details of the parameter space. I agree that given your relatively small sample size, there might be regions of the parameter space that you are disregarding. However, I suspect that it will be fairly minimal.

One reason for this statement is that other studies have applied HBV in the same region with better model performance (Beck et al., 2016, http://onlinelibrary.wiley.com/doi/10.1002/2015WR018247/epdf). Although, indeed, somewhat lower performance can be found in this study as well in the great-plains region (Figure 2). Another explanation, besides the parameter sample, could be the warm-up period as discussed above.

* Section 4: "Climate models disagree more in a more extreme scenario" - How much of this can be attributed simply to the disconnect between your models and the climate models themselves? (See issue regarding offline/online models above)

Based on our results, we cannot say how much of the disagreement could be decreased in an online application, although that would be very interesting. However, the point that we were trying to raise is that, with more extreme emission scenarios,

the spread in projections among the climate models increases. This is regardless of whether this projection is subsequently applied to an offline hydrologic model. Question remains, indeed, whether in an online application the spread would decrease.

* "Furthermore, land-use and soil parameters have been kept constant for both modelling periods. Although it is very likely that land-use will change in the future as a result of climate change or population growth, there are currently no methods to quantify this change and translate that to parameter values for the employed hydrologic models." -> GCMs actually do currently account for land use change. Now to be fair, these changes are mostly due to offline studies that predict that land use will be in the future, however, this information is out there and could be used in theory within the hydrologic models.

Yes, there is literature available on land-use change projections, but it is not straightforward to translate these changes to the changes in the parameters of our conceptual hydrologic models (for VIC it would be fairly easy because land-use is explicitly parameterized, but SAC and HBV are more schematic). We will, however, include some references to land-use change studies for the interested reader.

---

## Author Response (AR1)

**Rebuttal for**
**"Mapping (dis)agreement in hydrologic projections"**
**(hess-2017-564)**

We would like to thank the reviewers for their positive response and constructive feedback. We have incorporated most suggestions from the reviewers. Below you can find a point-by-point discussion of the reviewer comments.

**Reviewer 1:**

*p.2 l.26: I suggest to mention the information about which RCP you used in chapter 2.2*

RCP8.5 was mentioned in section 2.2, but we have mentioned it separately now to give it more emphasis (p.6, l.6)

*p.3 Fig 1d is not clear to me: why two bars in the upper positive change section? What is the black line with the two points and how is it created? Why is 'Frequency' written below the box?*

The changes from the six points compared to the 1:1 line in Figure 1c are summarized in a histogram as shown in figure 1d. This histogram is turned 90degrees, and therefore frequency is shown on the x-axis. The black line shows the mean change from the distance from the six points in 1c to the 1:1 line. We have adapted the description in the caption and added the word 'histogram' to panel *d* to make this more clear.

*p.4 l.1ff: Model description: I miss one sentence for explaining the model concept about runoff components (surface, lateral, groundwater) which are important for discharge timing*

A description of the different runoff mechanisms in the models has been added (see Section 2.2).

*p.4 l.3: moisture p.5 l.1: what are the "100 base runs with average parameter values" - how were these defined? e.g. nothing about this is mentioned in the parameter tables (Annexe)*

The 100 base runs were based on the mean parameter value (i.e., in the middle between the min and the max boundary as mentioned in the table). The other part of the sample consists of perturbations of this base sample. This explanation has been added to the text (p.5, l.12-14).

*p.5 l.4: I miss one or two sentences about the other input data. I know it is publicly available in the CAMELS dataset. But I think it is important to know the very basics here: Topography, land use, soil/geology and if catchment management (irrigation, damming) is considered and included in the dataset and in models. If catchment management is not included, could that be a reason for the non-behavioural catchments? I.e. the central US is subject to the highest ratio of agriculture and this distribution seems to fit well to the non-behavioural spots*

We have added a section (2.1) that briefly describes the catchments in our dataset. As explained there, the catchments have been selected such that human influence is minimal. Therefore, we think that this cannot (at least not directly) explain the non-behavioural catchments in the central US.

*p.5 l.16 and p.6 l.2: Please comment on why the two time periods differ in length. When comparing aggregated/average metrics the length of the comparison period is important since the longer the time series, the less influenced are the metrics by singular annual extremes.*

The two periods differ in length because the available observations only covered a 28-year period (of which we consider 23 years due to the 5 year model spin-up), while generally 30 years is considered representative for climate. So, we have decided for the future period to obey the 30 year period. Indeed this results in two different lengths which might have some influence on the calculated mean, but we think this effect is very limited.

*p.5 l.20: what is meant by "member" (i.e. what distinguishes the different 'members' of each GCM family - regional climate model, version, resolution, year, ....)?*

Knutti et al (2013) defined GCM families based on their output: "based on the predicted change in temperature and precipitation fields for the end of the 21st century in the RCP8.5 scenario relative to the control." (Figure 1 in Knutti et al., 2013). By selecting one member (GCM) of each GCM-family, we approach the full range of projections by all GCMs. This explanation has been added to the text (p.6, l.6-9).

*p.6 l.6-7: I suggest to define what you mean by "ensemble mean change". E.g. I think something along these lines is clearer: "The ensemble mean change was then determined as the mean change over all behavioural parameter sets of each GCM-, hydrological model-, and catchment combination".*

The ensemble mean change has only been determined over all parameter sets. When different hydrological models or GCMs are compared, we only compared the ensemble mean change mutually, as is explained in the next paragraph. We have added a sentence to make this clearer (p. 6, l.30).

*p.6 l.12: I assume number of "representative sample of parameter sets" is defined through the behavioural runs. For the other two uncertainty sources we know the number (three for the hydro models, five for the GCMs) - but for the chosen parameter sets you do not show them. However, I think it matters how many runs in each catchment are used to produce all the subsequent results. Could you show three additional maps of the CONUS (could also go to the Appendix) where the color of each catchment dot indicates the number of behavioural runs for each hydrologic model?*

The three maps have been added in the supplementary material (Figure C1), and are referred to in the methods section.

*p.6 l.18-26: I really like this part of the analysis, but the paragraph is difficult to understand without having seen the results and I suggest to begin the paragraph with an explanation, e.g. something along the lines: "It is assumed that catchment characteristics can influence the agreement between hydrological models and GCM. To assess the influence on the hydrological model agreement, we divided all basins into three categories: ... "*

The suggestion has been added to the text (p.7, l.9-11).

*p.8 l.7-8: Is this result not better suited for the section 3.2.1?*

We think not, because these results specifically refer to parameter-disagreement, for a specific model. We admit, however, that this can be confusing, and we added a sentence to explain this (p.9, l.14).

*p.8 l.18ff: I think this is an important paragraph which is valid for the other sources of uncertainty as well. I.e. if you end up with model runs that generally depict the processes better, you may end up with less disagreement for the other uncertainty causes as well. So, an interesting hypothesis to test would be, if most of your mapped disagreement is caused by parameter sets at the lower end of the KGE and if high KGEs lead to higher agreement (though I am interested in this, this is just a side note, no need to do this within this paper). However, I think the paragraph fits better to the discussion and I suggest to add: - that it could be possible that improved process depiction in your models could reduce disagreement related to other uncertainty sources as well - numerous studies (e.g. Pool et al. 2017 HESS 21) have found that looking at certain metrics without having used them in the optimization (in your case: selection of behavioural runs) can cause inadequate*

*depiction of those metrics - so the actual selection of the objective function may influence (dis-)agreement - a short statement if you can rule out that the non-behavioural results could be due to the selection of the parameters and ranges which may be more suitable for conditions significantly different from the catchments that are non-behavioural*

This is a valuable remark that we therefore have added to the discussion of these results (p.9, l.26). We did not incorporate the suggestion to move this section to the Discussion, because in the structure of the manuscript we decided to make the Discussion-section more general, and do the detailed discussion of the results within the results section.

*p.11. l.29: a lower aridity? Figure 5.d. suggests higher aridity? how can aridity be both high for disagreement and non-behavioural catchments?*

Thank you, it should indeed be higher aridity, the figure has been adapted.

*also at Figure 7.d: the significance triangle for mean delta P should point down and be hollow or? Seems C4 like I have difficulties understanding the rose plots. If the plots are correct, I require more explanation how they need to be interpreted (e.g. already in the methods with an example rose plot).*

This was indeed incorrect, thank you. Figure 7 and 8 have been adapted.

*p.15 l.10: This chapter is a very good summary of the uncertainties. But I miss that you explain how the combined uncertainty is produced in the methods. Did I miss something?*

This was indeed not clear in the Methods-section. Therefore, we have added a section that describes the procedure (Section 2.6).

*I suggest to add a few sentences which hydrologic model, which GCM and which combination led to the highest (dis)agreement. I know that the information is scattered throughout the results, but I would have liked to see this information summarized in the conclusions. I envision these sentences as a very concise summary of the whole Appendix.*

It is not straightforward to provide the information on which hydrologic model and GCM combination led to the highest agreement since in order to do this we have to compare mutually (i.e., we cannot say VIC had the highest agreement because to say something about agreement among hydrologic models we compared VIC with SAC and HBV). Furthermore, besides it practically being not possible to directly provide this information, we are afraid that it can give a false sense of certainty because agreement still does not imply that the projection is *true*. Therefore, we have decided not to add this information to the conclusion.

**Reviewer 2:**

*I was very impressed with the depth of the introduction, methods, and results. However, I was disappointed by the discussion section. I believe that there are many things that could (and should!) be discussed regarding the implications of this work that would be a missed opportunity to not include in the discussion section. For example, what is the physical explanation (if any) of why you get the results that you do? There is plenty of material in the results to enable this discussion and I believe it would be quite useful. Another topic of discussion is also another main comment. One of the underlying assumptions that is made in this and all offline studies that use GCM output as input to an offline hydrologic model is that the differences in hydrologic projections based on using different model parameters and hydrologic models does not impact the climate system (i.e., feedbacks). In other words, the precipitation and temperature that you are using from the GCM output depend on a land surface model that is contained within its original GCM; changing the model and parameters will directly change the temperature and precipitation you get... As a result, the most complete way*

*to approach this type of study would be to use your approach in coupled GCMs. However, I completely understand why this is computationally not feasible when you are looking at all the different sources of uncertainty. That being said, I believe it is important to at least discuss this problem and mention how you believe the results might change if it were feasible to do this study "online" instead of "offline". See the following paper for more background on this issue: "Milly, P. C., & Dunne, K. A. (2016). Potential evapotranspiration and continental drying. Nature Climate Change, 6(10), 946-949"*

We thank the reviewer for this comment. Based on the suggestion from reviewer 1 and reviewer 2, we have expanded the discussion substantially (see p. 19), including possible physical explanations for our results and the discussion of the offline application.

*Was the GCM output downscaled and bias corrected against the forcing used in the 1980-2008 simulations? I understand that it has indeed been bias corrected; however, bias correction is always done against some reference database; I would hope that that reference database is the one being used to force the 1980-2008 simulations. Please clarify. If it was not bias corrected against the forcings used for the 1980-2008 simulations, I am concerned that some of your signal in change in annual mean runoff could be attributed simply to discrepancies between the observed forcing and the "biascorrected" GCM output.*

The GCM output was bias corrected on the Maurer-dataset for the period 1950-1999, so this is a different dataset than where the 1980-2008 reference simulations have been performed with (which was the Daymet-dataset). We compared the temperature and precipitation differences between Daymet and Maurer for the 605 basins. Generally, both data-sets agree pretty well (similar in the mean), although precipitation differences exist in the west, and there are temperature differences in the central US and in the higher basins.

We expected that the effect of the difference in calibration-dataset and bias-correction dataset on the conclusion of our study is limited, since we do not compare the Daymet- forced model results with the GCM forced model results, but the two GCM-forced model results (historical versus future). So the effect of this bias correction only applies to the calibration. We re-ran all basins, this time forced with Maurer input data, and investigated the differences. Tables 1 and 2 provide an overview of the results for the calibration when looking at the mean runoff only. When investigating the behavioural runs in more detail, both forcing-datasets lead to a comparable number of behavioural runs (on average 2, 2, and 1% difference in number of behavioural runs for VIC, SAC and HBV, respectively, Table 1). However, the number of basins with disagreement among the behavioural parameter sets differed on average 11% between the two forcing datasets, with Maurer generally leading to a higher number of basins with parameter disagreement. So, calibration based on Maurer-forcing leads, on average, to a slightly higher number of basins experiencing parameter disagreement. Overall, using a different forcing dataset did impact the calibration but we do think however, that the results presented in our manuscript remain valid.

Table 1: Difference in average percentage of runs over 605 basins that was classified as behavioural.

| VIC | 2% |
|-----|-----|
| SAC | 2% |
| HBV | 1% |

Table 2: Percentage of basins where the agreement among the bahavioural parameter runs is different between the Maurer-forced calibration and Daymet-forced calibration.

|     | cnrm | ipsl | ccsm | mpi | inmcm |
|-----|------|------|------|-----|-------|
| VIC | 6    | 10   | 7    | 10  | 8     |
| SAC | 15   | 17   | 8    | 13  | 19    |
| HBV | 13   | 9    | 7    | 12  | 14    |

*Using 5 years for spin-up is awfully low. Are you sure that this is appropriate? I would have suggested cycling through the 1980 to 2008 a few times. Although this most likely does not disqualify the results, there should be some argument in the paper for why only 5 years appeared to be enough.*

For the most complex hydrologic model included in this study, VIC, when run on an hourly basis, three months was shown to be sufficient (Melsen et al., 2016, [https://doi.org/10.5194/hess-20-2207-2016](https://doi.org/10.5194/hess-20-2207-2016)). For HBV, it has been stated that one year warm-up period on a daily basis is sufficient in most cases (Seibert and Vis, 2012; [www.hydrol-earth-syst-sci.net/16/3315/2012/](www.hydrol-earth-syst-sci.net/16/3315/2012/)). Therefore we assumed five years to be ample. An important critical note here is indeed that the warm-up period can be longer in drier climates, which could perhaps explain the lower model performance in these regions. These remarks have been added to the manuscript (p.5, l. 19-21, p.9, l.29-32).

*Section 3.1.2: "A larger parameter sample could therefore decrease the number of non-behavioural basins and even allow for a more stringent selection criterion." Maybe, but certainly not necessarily. Assuming that the LHS sample is robust, then your model parameters will already capture the details of the parameter space. I agree that given your relatively small sample size, there might be regions of the parameter space that you are disregarding. However, I suspect that it will be fairly minimal.*

One reason for this statement is that other studies have applied HBV in the same region with better model performance.  Another explanation, besides the parameter sample, could be the warm-up period as discussed above. These remarks have been added to the manuscript (p.9, l.29-32)

*Section 4: "Climate models disagree more in a more extreme scenario" - How much of this can be attributed simply to the disconnect between your models and the climate models themselves? (See issue regarding offline/online models above)*

This is indeed a relevant question, that we cannot answer based on our results. We have removed this statement from the discussion.

*"Furthermore, land-use and soil parameters have been kept constant for both modelling periods. Although it is very likely that land-use will change in the future as a result of climate change or population growth, there are currently no methods to quantify this change and translate that to parameter values for the employed hydrologic models." -> GCMs actually do currently account for land use change. Now to be fair, these changes are mostly due to offline studies that predict that land use will be in the future, however, this information is out there and could be used in theory within the hydrologic models.*

We have adapted the sentences, to indeed make clear that the GCMs do partially  account for land use change which implies that this information is available. We also explain why we could not implement the changes in this study (p. 20, l. 11-14).

We believe that, with the implemented changes, the manuscript has become more complete.

On behalf of all co-authors,

Lieke Melsen

---

## Author Response (AR2)

Dear editor,

Thank you for your positive response to our review and for the useful suggestion. Here below you can find a short response. All changes are indicated in red in the manuscript.

*- In the new Section 2.1, I think that it is worth adding a sentence about the record lengths of the 605 basins to address comment #6 of Reviewer 1.*

We have added a sentence to make this clear; see p.4, line 2-5.

*- In Section 2.2, spell out the various models and their abbreviation for the Section headings. For example, the section heading for VIC should read: "Variable Infiltration Capacity (VIC)".*

The headings have been adapted.

*- Please ensure that the manuscript complies with HESS manuscript guidelines found here: https://www.hydrology-and-earth-system-sciences.net/for_authors/manuscript_preparation.html*

All abbreviations in the figures have been spelled out in their caption, for most figures this implied that the abbreviations that are used for the hydrologic models had to be spelled out. Furthermore, KGE has been replaced with a subscript-notation.

*- The use of North West or North East (for example on p. 19, lines 4 and 15) are awkward. I would revise to read "northwest United States" and northeast United States". The current reading makes these terms seem like an absolute direction, as in the "North Pole".*

We agree and have adapted this formulation in the Discussion section.

Kind regards,

On behalf of all co-authors,
Lieke Melsen